# VISION-LANGUAGE MODELS PROVIDE PROMPTABLE REPRESENTATIONS FOR REINFORCEMENT LEARNING

## ABSTRACT

Intelligent beings have the ability to quickly learn new behaviors and tasks by leveraging background world knowledge. This stands in contrast to most agents trained with reinforcement learning (RL), which typically learn behaviors from scratch. Therefore, we would like to endow RL agents with a similar ability to leverage contextual prior information. To this end, we propose a novel approach that uses the vast amounts of general-purpose, diverse, and indexable world knowledge encoded in vision-language models (VLMs) pre-trained on Internet-scale data to generate text in response to images and prompts. We initialize RL policies with VLMs by using such models as sources of *promptable representations*: embeddings that are grounded in visual observations and encode semantic features based on the VLM's internal knowledge, as elicited through prompts that provide task context and auxiliary information. We evaluate our approach on visually-complex RL tasks in Minecraft. We find that policies trained on promptable embeddings extracted from general-purpose VLMs significantly outperform equivalent policies trained on generic, non-promptable image encoder features. Moreover, we show that these representations outperform instruction-following methods and are competitive with domain-specific representations. In ablations, we find that VLM promptability and text generation both are important in yielding good representations for RL. Finally, we give a simple method for evaluating and optimizing prompts used by our approach for a given task without running expensive RL trials, ensuring that it extracts task-relevant semantic features from the VLM.

## 1 INTRODUCTION

Embodied decision-making often requires representations informed by extensive world knowledge for perceptual grounding, planning, and control. Humans can rapidly learn to perform sensorimotor tasks by drawing on prior knowledge, which might be high-level and abstract ("If I'm cooking something that needs milk, the milk is probably in the refrigerator") or grounded and low-level (e.g., what refrigerators and milk look like). These capabilities would prove highly beneficial for reinforcement learning (RL) too: we aim for our agents to interpret tasks in terms of concepts that can be reasoned about with relevant prior knowledge and grounded with previously-learned representations, thus enabling more efficient learning. However, doing so requires a condensed source of vast amounts of general-purpose world knowledge, captured in a form that allows us to specifically index into and access *task-relevant* information. Therefore, we need representations that are contextual, such that agents can use a concise task context to draw out relevant background knowledge, abstractions, and grounded features that aid it in acquiring a new behavior.

An approach to facilitate this involves integrating RL agents with the prior knowledge and reasoning abilities of pre-trained foundation models. Transformer-based language models (LMs) and vision-language models (VLMs) are trained on Internet-scale data to enable generalization in downstream tasks requiring facts or common sense. Moreover, in-context learning (Brown et al., 2020) and instruction fine-tuning (Ouyang et al., 2022) have provided better ways to index into (V)LMs' knowledge and guide their capabilities based on user needs. These successes have seen some transfer to embodied control, with (V)LMs being used to reason about goals to produce executable plans (Ahn et al., 2022) or as pre-trained encoders of useful information (like instructions (Liu et al., 2023) or feedback (Sharma et al., 2023)) that the control policy utilizes. Both of these paradigms have major

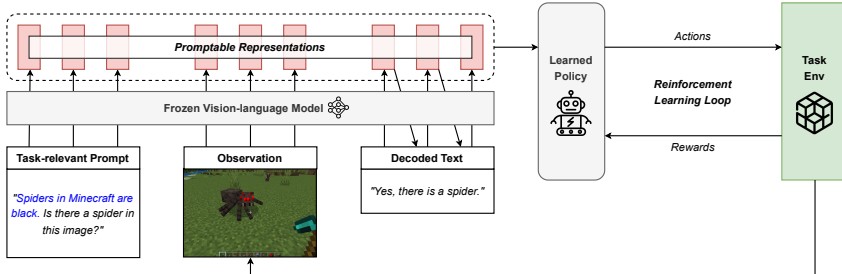

Figure 1: **An example instantiation of PR2L.** We query a VLM with a *task-relevant prompt* about observations to produce *promptable representations*, which we train a policy on via RL. Rather than directly asking for actions or specifying the task, the prompt enables *indexing into the VLM's prior world knowledge* to access task-relevant information. They also allow us to inject auxiliary information (e.g., about visual features).

limitations: actions generated by LMs are often not appropriately grounded, unless the tasks and scenes are amenable to being expressed or captioned in language. Even then, (V)LMs are often only suited to producing subtask plans, not low-level control signals. On the other hand, using (V)LMs to simply encode inputs under-utilizes their knowledge and reasoning abilities, instead focusing on producing embeddings which reflect language's compositionality (e.g., so an instruction-following policy may generalize). This motivates the development of an algorithm for learning to produce low-level actions that are both grounded and that leverage (V)LMs' knowledge and reasoning.

To this end, we introduce **P**romptable **R**epresentations for **R**einforcement **L**earning (**PR2L**): a flexible framework for guiding vision-language models to produce *semantic features*, which (i) integrate observations with prior task knowledge, and (ii) are grounded into actions via RL (see Figure 1). Specifically, we ask a VLM questions about observations that are related to the given control task, encouraging it to attend to task-relevant information in the image based on both its internal world knowledge and any supplemental information injected via prompting. The VLM then encodes this information in decoded tokens, which are discarded, and associated representations, which serve as input to a learnable policy. In contrast to the standard approach of using pre-trained image encoders to convert visual inputs into *generic* features for downstream learning, our method yields *task-specific* features that capture information particularly conducive to learning a considered task. In this way, the VLM does not just produce an ungrounded encoding of instructions or task specifications, but embeddings containing semantic information relevant to the given task that is both grounded and informed by the VLM's prior knowledge through prompting.

To the best our knowledge, we introduce the first approach for initializing RL policies with generative VLM representations. We demonstrate our approach on tasks in Minecraft (Fan et al., 2022), as it has semantically-rich and visually-complex tasks found in many practical, realistic, and challenging applications of RL. We find that, by using our approach, we outperform equivalent policies trained on unpromptable visual embeddings or with instruction-conditioning– both popular ways of using pre-trained image models and VLMs respectively for control. Furthermore, we show that promptable representations extracted from general-purpose VLMs outperform domain-specific representations. Our results and ablations highlight how visually-complex control tasks can benefit from accessing the knowledge captured within VLMs via prompting.

## 2 RELATED WORKS

**Embodied (V)LM reasoning.** Many recent works have leveraged (V)LMs as embodied reasoners by treating them as priors over effective plans for a given goal. These works use the model's language modeling and auto-regressive generation capabilities to extract such priors as textual subtask sequences (Ahn et al., 2022; Huang et al., 2022b; Sharma et al., 2022) or code (Liang et al., 2023; Singh et al., 2022; Zeng et al., 2022; Vemprala et al., 2023), by effectively using the LM to decompose long-horizon tasks into executable parts or instructions. These systems often need grounding mechanisms to ensure feasibility of their plans (e.g., affordance estimators (Ahn et al., 2022), scene captioners (Zeng et al., 2022), or trajectory labelers (Palo et al., 2023)). Furthermore, these works often assume access to low-level policies that can execute these subtasks, such as skills to allow a robot to pick up objects (Ahn et al., 2022; Liang et al., 2023), which is often a strong assumption. These methods generally do not address how such policies can be acquired, nor how these low-level

skills can themselves benefit from the prior knowledge in (V)LMs. Even works in this area that use RL still use (V)LMs as state-dependent priors over reasonable high-level goals to learn (Du et al., 2023). This is a key difference from our work: instead of considering priors on plans or goals, we rely on VLM's implicit knowledge *of the world* to extract representations which encode task-relevant information. We train a policy to solve the task by converting these features into low-level actions via standard RL, meaning the VLM does not need to know how to take actions for a task.

**Embodied (V)LM pre-training.** Other works use (V)LMs to embed useful information like instructions (Liu et al., 2023; Myers et al., 2023; Lynch & Sermanet, 2021; Mees et al., 2023), feedback (Sharma et al., 2023; Bucker et al., 2022), reward specifications (Fan et al., 2022), and data for world modeling (Lin et al., 2023b; Narasimhan et al., 2018). These works use (V)LMs as *encoders* that capture the compositional semantic structure of input text and images, which often aids in generalization: a instruction-conditioned model may never have learned to grasp apples (but was trained to grasp other objects), but by interacting with them in other ways and receiving associated language descriptions, the model might learn what an apple is and its physical properties, thus potentially being able to grasp it zero-shot. In contrast, our method's primary advantage is that the resulting embeddings are informed by world knowledge, both from prompting and pretraining. Rather than just specifying that the task is to acquire an apple, we ask a VLM to parse observations into directly relevant features, like whether there is an apple in the image or if the observed location is likely to contain apples – all information that is useful for RL, even in single-task settings. Thus, we use VLMs to help RL solve new tasks, rather than just to learn how to perform instruction following.

We note these two categories are not binary. For instance, Brohan et al. (2023) use VLMs to understand instructions, but also reasoning (e.g., figuring out the "correct bowl" for a strawberry is one that contains fruits); Palo et al. (2023) use a LM to reason about goal subtasks and a VLM to understand when a trajectory matches a subtask description, automating the demonstration collection/labeling of Ahn et al. (2022), while Adeniji et al. (2023) use a similar framework to pretrain a language-conditioned RL policy that can then be transferred to learning other tasks; and Shridhar et al. (2021) use CLIP to merge vision and text instructions directly into a form that a Transporter (Zeng et al., 2020) policy can operationalize. Nevertheless, these works primarily focus on instruction following in robot manipulation domains. In contrast, our approach prompts a VLM to supplement RL with representations of world knowledge, rather than relying on commands or task specifications. In addition, except for Adeniji et al. (2023), these works focus on imitation learning, assuming access to existing demonstrations for policy training and fine-tuning, which we forgo by using online RL.

## 3 PRELIMINARIES

**Reinforcement learning task and objective.** We adopt the standard deep RL partially-observed Markov decision process (POMDP) framework, wherein a given control task is defined by the tuple $(\mathcal{S}, \mathcal{A}, p_T, \gamma, r, \rho_0, \mathcal{O}, p_E)$, where $\mathcal{S}$ is the set of states, $\mathcal{A}$ is the set of actions, $p_T : \mathcal{S} \times \mathcal{A} \times \mathcal{S} \to \mathbb{R}$ are the state transition probabilities, $\gamma \in (0, 1)$ is the discount factor, $r : \mathcal{S} \times \mathcal{A} \to \mathbb{R}$ is the reward function, $\rho_0 : \mathcal{S} \to \mathbb{R}$ is the distribution over initial states, $\mathcal{O}$ is the set of observations (including the visual observations), and $p_E : \mathcal{S} \times \mathcal{O} \to \mathbb{R}$ are observation emission probabilities. The objective is to find parameters $\theta$ of policy $\pi_\theta : \mathcal{O} \times \mathcal{A} \to \mathbb{R}$ which, together with $\rho_0$, $p_E$, and $p_T$, defines a distribution over trajectories $p_\theta$ with maximum expected returns $\eta(\theta)$:

$$\eta(\theta) = \mathbb{E}_{((s_0,o_0,a_0),(s_1,o_1,a_1),...)\sim p_\theta} \left[ \sum_{t=0}^{\infty} \gamma^t r(s_t, a_t) \right] \tag{1}$$

**Vision-language models.** In this work, we utilize *generative VLMs* (like Li et al. (2022; 2023a); Dai et al. (2023)): models that generate language in response to an image and a text prompt passed as input. This is in contrast to other designs of combining vision and language that either generate images or segmentation (Rombach et al., 2022; Kirillov et al., 2023) and CLIP (Radford et al., 2021).

Formally, the VLM enables sampling from $p(x_{1:K}|I, c)$, where $x_{1:K}$ represents the $K$ tokens of the output, $I$ is the input image(s), $c$ is the prompt, and $p$ is the distribution over natural language responses produced by the VLM on those inputs. Typically, the VLM is pre-trained on tasks that require building association between vision and language such as image captioning, visual-question answering, or instruction-following. While these differ from the "pure" language modeling objective, all these tasks nonetheless require learning to attend to certain semantic features of input

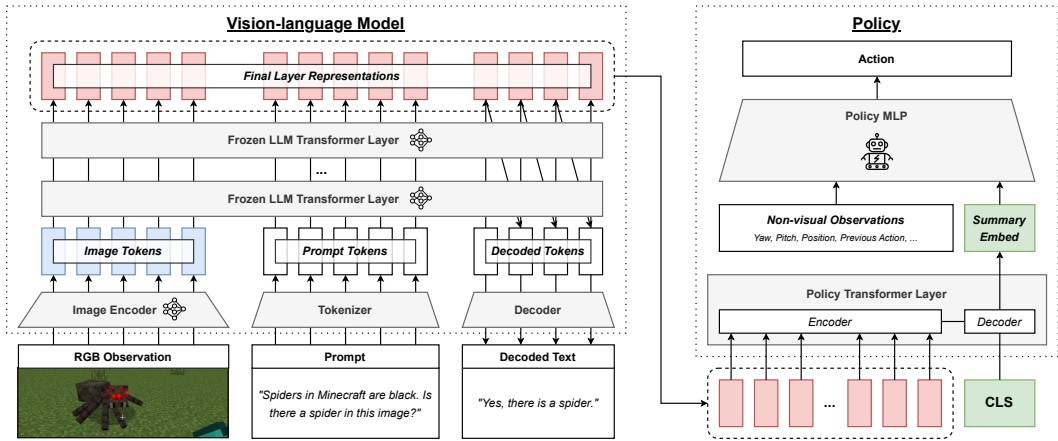

Figure 2: **Schematic of how we extract task-relevant features from the VLM and use those representations in a policy that we train with RL.** These representations can incorporate task context from the prompt, while generic image encoder representations cannot. As the embeddings of generative Transformers can be variable length, the policy also has a Transformer layer that takes in the VLM representations and a "CLS" token, thereby condensing all inputs into a single summary embedding.

images depending on the given prompt. For auto-regressive generative VLMs, this distribution is factorized as $\prod_t p(x_t|I, c, x_{1:t-1})$. Typical architectures for generative VLMs parameterize these distributions using weights that define a representation $\phi_t(I, c, x_{1:t-1})$, which depends on the image $I$, the prompt $c$, and the previously emitted tokens, and a decoder $p(x_t|\phi_t(I, c, x_{1:t-1}))$, which defines a distribution over the next token.

## 4 PR2L: PROMPTABLE REPRESENTATIONS FOR RL

Our goal is to supplement RL with task-relevant information extracted from VLMs containing general-purpose knowledge. One way to index into this information is by prompting the model to get it to produce semantic information relevant to a given control task. Therefore, our approach, PR2L, queries a VLM with a task-relevant prompt for each visual observation received by the agent, and receives both the decoded text and, critically, the intermediate representations, which we refer to as *promptable representations*. Even though the decoded text might often not be correct or directly usable for choosing the action, our key insight is that these VLM embeddings can still provide useful semantic features for training control policies via RL. This recipe enables us to incorporate semantic information without the need of re-training or fine-tuning a VLM to directly output actions, as proposed by Brohan et al. (2023). Note that our method is *not* an instruction-following method, and it does not require a description of the actual task in natural language. Instead, our approach still learns control via RL, while benefiting from the incorporation of *background context* to improve RL. In this section, we will describe various components of our approach, accompanied by design choices and other practical considerations.

### 4.1 PROMPTABLE REPRESENTATIONS

Why do we choose to use VLMs in this way, instead of the many other ways of using them for control? In principle, one can directly query a VLM to produce actions for a task given a visual observation. While this may work when high-level goals or subtasks are sufficient, VLMs are empirically bad at yielding the kinds of low-level actions used commonly in RL (Huang et al., 2022a). As VLMs are mainly trained to follow instructions and answer questions about visual aspects of images, it is more appropriate to use these models to extract *semantic features* about observations that are conducive to being linked to actions. Specifically, we elicit features that are useful for the downstream task by querying these VLMs with *task-relevant prompts* that provide contextual task information, thereby causing the VLM to attend to and interpret appropriate parts of observed images. Extracting these features naïvely by only using the VLM's *decoded text* has its own challenges: such models often suffer from both hallucinations (Ji et al., 2023) and an inability to report what they "know" in language, even when their embeddings contain such information (Kadavath et al., 2022; Hu & Levy, 2023). However, even when the text is bad, the underlying *representations* still contain valuable granular world information that is potentially lost in the projection to language

(Li et al., 2021; Wiedemann et al., 2019; Huang et al., 2023; Li et al., 2023b). Thus, we disregard the generated text in our approach and instead provide our policy the embeddings produced by the VLM in response to prompts asking about relevant semantic features in observations instead.

**Which parts of the network can be used as promptable representations?** The VLMs we consider are all based on the Transformer architecture (Vaswani et al., 2017), which treats the prompt, input image(s), and decoded text as token sequences. This architecture provides a source of learned representations by computing embeddings for each token at every layer based on the previous layer's token embeddings. In terms of the generative VLM formalism introduced prior, a Transformer-based VLM's representations $\phi_t(I, c, x_{1:t-1})$ consist of $N$ embeddings per token (the outputs of the $N$ self-attention layers) in the input image $I$, prompt $c$, and decoded text $x_{1:t-1}$. The decoder $p(x_t|\phi_t)$ extracts the final layer's embedding of the most recent token $x_{t-1}$, projecting it to a distribution over the token vocabulary and allowing for it to be sampled. When given a visual observation and task prompt, the tokens representing the prompt, image, and answer consequently encode task-relevant semantic information. Thus, for each observation, we use the VLM to sample a response to the task prompt $x_{1:K} \sim p(x_{1:K}|I, c)$. We then use some or all of these token embeddings $\phi_K(I, c, x_{1:t-1})$ as our promptable representations and feed them, along with any non-visual observation information, as a state representation into our downstream neural-network policy trained with RL.

In summary, our approach involves creating a task-relevant prompt that provides context and auxiliary information. This prompt, alongside the current visual observation from the environment, is fed to into the VLM to generate tokens. While these tokens are used for decoding, they are ultimately discarded. Instead, we utilize the *representations* produced by the VLM (associated with the image, prompt, and decoded text) as input for our policy, which is trained via an off-the-shelf online RL algorithm to produce appropriate actions. A schematic of our approach is depicted in Figure 2.

## 4.2 DESIGN CHOICES FOR INSTANTIATING PR2L

To instantiate our method, several design choices must be made. First, the representations of the VLM's decoded text are dependent on the chosen decoding scheme. E.g., greedy decoding is fast and deterministic, but may yield low-probability decoded tokens; beam search improves on this by considering multiple "branches" of decoded text, at the cost of requiring more compute time (for potentially small improvements); lastly, sampling-based decoding can quickly yield estimates of the maximum likelihood answer, but at the cost of introducing stochasticity, which may increase variance. Given the inherent high-variance of our tasks (due to sparse rewards and partial observability) and the computational expense of VLM decoding, we opt for greedy decoding.

Second, one must choose which VLM layers' embeddings to utilize in the policy. While theoretically, all layers of the VLM could be used, pre-trained Transformer models tend to encode valuable high-level semantic information in their later layers (Tenney et al., 2019; Jawahar et al., 2019). Thus, we opt to only feed the final two layers' representations into our policy. It's worth noting that unlike conventional fixed-dimensional state representations used in RL, these representation sequences are of variable length. To accommodate this, we incorporate an encoder-decoder Transformer layer in the policy. At each time step in a trajectory, this Transformer receives variable-length VLM representations, which are attended to and converted into a fixed-length summarization by the embeddings of a learned "CLS" token (Devlin et al., 2019) in the decoder (green in Figure 2). We also note that this policy can receive the observed image directly (e.g., after being tokenized and embedded by the image encoder), so as to not lose any visual information from being processed by the VLM. However, we choose not to do this in our experiments in order to more clearly isolate and demonstrate the usefulness of the VLM's representations in particular.

Finally, while it is possible to fine-tune the VLM for RL end-to-end with the policy, akin to what was proposed by Brohan et al. (2023), this approach incurs substantial compute, memory, and time overhead, particularly with larger VLMs. Nonetheless, we find that our approach performs better than not using the language and prompting components of the VLM. This holds true even when the VLM is frozen, and only the policy is trained via RL, or when the decoded text occasionally fails to answer the task-specific prompt correctly.

## 4.3 TASK-RELEVANT PROMPT

How do we design good prompts to elicit useful representations from VLMs? As we aim to extract good state representations from the VLM for a downstream policy, we do not use instructions or task

descriptions, but task-relevant prompts: questions that make the VLM attend to and encode semantic features in the image that are useful for the RL policy learning to solve the task. For example, if the task is to find a toilet within a house, appropriate prompts include "Is there a toilet in this image?" and "Am I likely to find a toilet here?" Intuitively, the answers to these questions help determine appropriate actions (approach the toilet, look around the room, explore elsewhere, etc.), making the corresponding representations good for representing the state for a policy. Answering the questions will require the VLM to attend to task-relevant features in the scene, relying on the model's internal conception of what things look like and common-sense semantic relations. Note that prompts based on instructions or task descriptions do not enjoy the above properties: while the goal of those prior methods is to be able to directly query the VLM for the optimal action, the goal of task-relevant prompts is to produce a useful state representation, such that running RL optimization on them can accelerate learning an optimal policy. While the former is not possible without task-specific training data for the VLM in the control task, the latter proves beneficial with off-the-shelf VLMs. Finally, these prompts also provide a place where auxiliary helpful information can be provided: for example, one can describe what certain entities of interest look like, aiding the VLM in detecting them even if they were not commonly found in the model's pre-training data.

**Evaluating and optimizing prompts for RL.** Since the specific information and representations elicited from the VLM are determined by the prompt, we want to design prompts that produce promptable representations that maximize performance on the downstream task. The brute-force approach would involve running RL with each candidate prompt to measure its efficacy, but this would be computationally very expensive. In lieu of this, we evaluate candidate prompts on a small dataset of observations labeled with semantic features of interest for the considered task. Example features include whether task-relevant entities are in the image, the relative position of said entities, or even actions (if expert demonstrations are available). We test prompts by querying the VLM and checking how well the resulting decoded text for each image matches ground truth labels. As this is only practical for small, discrete label spaces that are easily expressed in words, we also draw from probing literature (Shi et al., 2016; Belinkov & Glass, 2019) and see how well a small model can fit the VLM's embeddings to the labels, thus measuring how extractable said features are from the promptable representations (without memorization). While this approach does not directly optimize for task performance, it does act as a proxy that ensures a prompt's resulting representations encode certain semantic features which are helpful for the task.

## 5 EXPERIMENTAL EVALUATION

We wish to empirically show that one can prompt a VLM to elicit visually-grounded representations that aid in a downstream control task, thus bringing the benefits of Internet-scale VLM pre-training to RL. To this end, we design experiments to answer the following questions: **(1)** Can promptable representations obtained via task-specific prompts enable more performant and **sample-efficient** learning than those of pre-trained image encoders? **(2)** How does PR2L compare to approaches that directly "ask" the VLM to generate the best possible actions for a task specified in the prompt? **(3)** How well do representations obtained from a general-purpose VLM compare to other domain-specific representations, that are also trained to associate visual observations with text, measured via control performance?

**Implementation details.** For all experiments, we use the InstructBLIP instruction-tuned generative VLM (Dai et al., 2023). Concretely, we use the Vicuna-7B version (Chiang et al., 2023) at half precision to produce promptable representations. We present the hyperparameters in Appendix C.

### 5.1 EXPERIMENTAL SETUP AND COMPARISONS: MINECRAFT

To answer the questions listed above, we conduct experiments on the Minecraft domain, which provides a number of control tasks that require associating visual observations with rich semantic information to succeed. Moreover, since these observations are distinct from the images in the the pre-training dataset of the VLM, succeeding on these tasks relies crucially on the efficacy of the task-specific prompt in meaningfully affecting the learned representation, enabling us to stress-test our method. For example, while spiders in Minecraft somewhat resemble real-life spiders, they actually exhibit stylistic exaggerations, such as bright red eyes and a large black body. If the task-specific prompt is indeed effective in informing the VLM of these facts, it would produce a representation that is more conducive to policy learning and this would be reflected in task performance. Finally, recent works also find pre-trained models to be useful in Minecraft tasks (Baker et al., 2022; Zhu

| | PR2L Prompt | RT-2-style Baseline Prompt | Change Auxiliary Text Ablation Prompt |
|---|---|---|---|
| *Combat Spider* | "Spiders in Minecraft are black. Is there a spider in this image?" | "I want to fight a spider. I can attack, move, or turn. What should I do?" | "Is there a spider in this image?" |
| *Milk Cow* | "Is there a cow in this image?" | "I want to milk a cow. I can use my bucket, move, or turn. What should I do?" | "Cows in Minecraft are black and white. Is there a cow in this image?" |
| *Shear Sheep* | "Is there a sheep in this image?" | "I want to shear a sheep. I can use my shears, move, or turn. What should I do?" | "Sheep in Minecraft are usually white. Is there a sheep in this image?" |

Table 1: Prompts used for querying the VLM with PR2L, comparison (b), and the change auxiliary text ablation. For the last column, we remove the auxiliary text for *combat spider*, and add it in for the other two.

et al., 2023; Nottingham et al., 2023; Lifshitz et al., 2023; Wang et al., 2023b), motivating us further to study this domain.

**Minecraft tasks.** We consider three Minecraft tasks provided by the MineDojo simulator (Fan et al., 2022): (i) ***combat spider***, where the agent must find and defeat a nearby spider while equipped with a shield, diamond sword, and diamond armor; (ii) ***milk cow***, where the agent must milk a nearby cow by using an equipped bucket; and (iii) ***shear sheep***, where the agent must cut wool from a nearby sheep by using equipped shears. We follow the prescriptions of Fan et al. (2022) for defining the observation and action spaces and reward function structures for these tasks. Specifically, at each time step, the policy observes an egocentric RGB image, its pose (Cartesian coordinates and pitch/yaw angles in the world frame), and its previously-executed action; the policy can choose a discrete action to turn the agent by changing the agent's pitch and/or yaw in fixed discrete increments, move, attack, or use a held item. The maximum allowed rollout length for each task is 500, with termination for early successes. We utilize proximal policy optimization (PPO) (Schulman et al., 2017) as our base RL algorithm for all approaches. Additional details are available in Appendix B.

**Comparisons.** To answer the research questions posed at the start of this section, we compare our approach to: **(a)** methods that do not utilize prompting to obtain representations of the observation, **(b)** a method that directly "asks" the VLM to output the action to execute on the agent, inspired by the approach of Brohan et al. (2023), and **(c)** running RL on the MineCLIP representation (Fan et al., 2022), which is obtained by fine-tuning CLIP (Radford et al., 2021) on Minecraft data. Running RL on MineCLIP serves as an "oracle" comparison since this representation was explicitly fine-tuned on a large dataset of Minecraft Youtube videos, whereas our pre-trained VLM is frozen, and is not trained on any Minecraft video data. Comparison (b) attempts to adapt the approach of Brohan et al. (2023) to our setting and directly outputs the action from the VLM. While Brohan et al. (2023) also fine-tune the VLM backbone, we are unable to fine-tune this VLM using our computational resources. In order to compensate for this difference, we do not just execute the action from the VLM, but train an RL policy to map this decoded output action into a better action. Note that, if the VLM already decodes good action texts for the specified task, then simply copying over this action via RL should be easy. Finally, comparison (a) does not utilize the task-specific prompt altogether, instead using embeddings from the VLM's image encoder (specifically, the ViT-g/14 image encoder from InstructBLIP (blue in Figure 2)). While this representation of the observation is task-agnostic and still benefits from pre-training, PR2L utilizes prompting to produce *task-specific* representations. For a fair comparison, we utilize the *exact same* Transformer-layer policy architecture and hyperparameters for this baseline as in PR2L, ensuring that performance differences come from prompting for better representations from the VLM. For more details, see Appendix C.

## 5.2 DESIGNING TASK-SPECIFIC PROMPTS FOR PR2L

Next, we discuss how to design the task-specific prompts for PR2L. As noted earlier in Section 4.3, these are not instructions or task descriptions, but prompts that force the VLM to encode semantic information about the task in its representation. The simplest relevant semantic feature for our tasks is the presence of the target entity in a given visual observation. Thus, we choose "Is there a [spider/cow/sheep] in this image?" as the base of our chosen prompt. We also introduce two alternative prompts per task that prepend different amounts of auxiliary information about the target entity. For example, for spiders, the two candidate prompts that we consider are "Spiders in Minecraft are black." and "Spiders in Minecraft are black and have red eyes and long, thin legs." To choose between these prompts, we apply our prompt evaluation and optimization strategy by measuring how well the VLM is able to decode a correct answer to the question of whether or not the target entity is present in the image on a small pre-collected dataset of images annotated with the answer to this question. Full details and results of this evaluation scheme are presented in Appendix A and Table

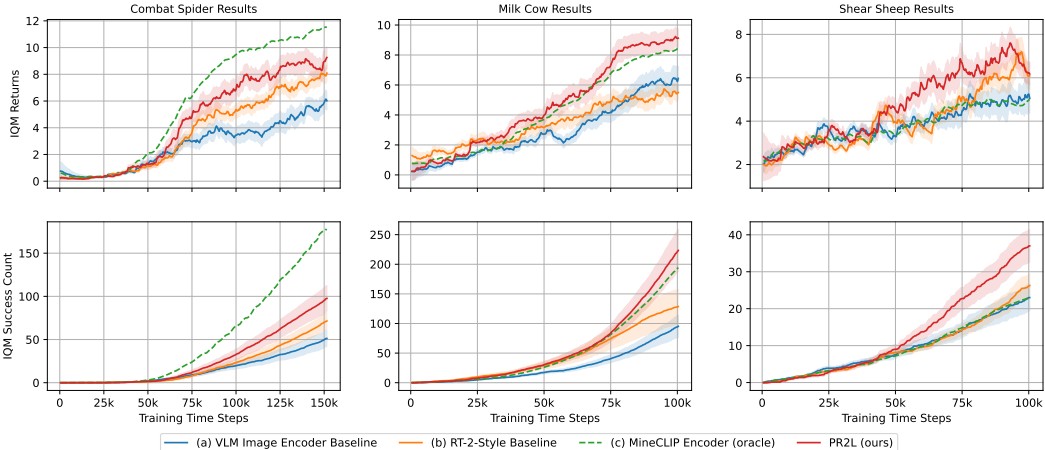

Figure 3: **Performance of PR2L vs other comparisons**. Plots show IQM returns and success counts over time for the Minecraft tasks for 16 trials. Shaded regions represent one standard error **(though some are omitted for visual clarity)**. PR2L outperforms the VLM image encoder and RT-2-style baselines, while being competitive with the domain-specific representations produced by the MineCLIP encoder oracle.

2. While we observe that auxiliary text only helps with detecting spiders while systematically and significantly degrading the detection of the other two entities, more importantly, our results (that we discuss in the next section) show that this detection success rate is correlated with performance of the RL policy for any prompt. Finally, we note that we also utilize additional prompts for comparison (b), following the recipe for prompt design prescribed by Brohan et al. (2023), which motivated this comparison. In these prompts, we also provide a list of allowed actions that the VLM can choose from to the policy. All chosen prompts are presented in Table 1.

## 5.3 RESULTS AND ABLATIONS

For all of our results, we report the interquartile mean (IQM) standard error of the returns and number of successes over 16 seeds per condition for all Minecraft tasks in Figure 3 and the probability of improvement of PR2L over the VLM image encoder baseline in Figure 4, following Agarwal et al. (2022). For the returns, we apply exponential smoothing to the

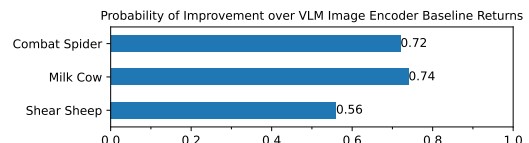

Figure 4: PR2L yields high probability of improvement over the VLM image encoder comparison (a).

episode's returns or success indicators with smoothing factor $\alpha = 0.05$ (so each smoothed datapoint is determined by an effective window of the 20 most recent unsmoothed ones).

**Minecraft results.** As shown in Figure 3, on all the three tasks, PR2L significantly outperforms both using the VLM image encoder (comparison (a)) and the method that directly "asks" the VLM for the action (comparison (b)) inspired by RT-2. This shows how control tasks can benefit from extracting prior knowledge encoded in VLMs by prompting them with task context and auxiliary information, even in single-task situations where the generalization properties of instruction-following methods do not apply. While PR2L does not outperform the "oracle" MineCLIP policy on *combat spider*, it performs competitively or better than MineCLIP on the other two tasks that we study, even though the latter is fine-tuned on Minecraft-specific data while InstructBLIP is not.

Furthermore, we hypothesize that MineCLIP outperforms PR2L on the spider task because, out of all the entities that we study, Minecraft spiders are the most different visually from real spiders, giving rise to comparatively poor representations in the VLM (which is trained exclusively on natural images). Nevertheless, our results in Figure 3 show that PR2L provides an effective approach to transform a general-purpose VLM into a strong task-specific control policy (without fine-tuning) that can often outperform policies trained on domain-specific representations on a given task.

**Ablations studies.** We run several ablation experiments to isolate and understand the importance of various components of PR2L towards extracting good promptable representations for RL. First, we run PR2L with *no prompt* to see if prompting with task context actually tailors the VLM's generated representations favorably towards the target task, improving over an unprompted VLM. Note that

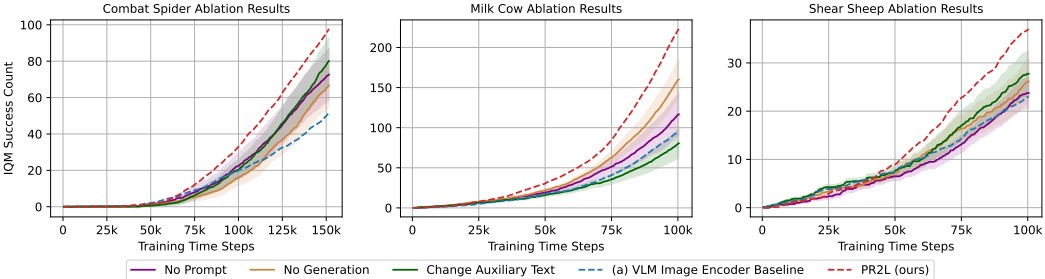

Figure 5: **Ablation studies on all Minecraft tasks** with the VLM image encoder baseline (blue) and our full approach (red), as shown in Figure 3. All ablations achieve worse performance than PR2L, highlighting the importance of each ablated component (the prompt, VLM generation, or inclusion of auxiliary text). Curves are IQM success counts and shaded regions are the standard error. We apply a third-order Savitsky-Golay filter with window size 10 to improve readability. We present additional metrics in Figure. 7 in the Appendix.

this is not the same as simply utilizing the image encoder (comparison (a)) alone, since this ablation decodes through the VLM, just with an empty prompt. Second, we run PR2L with our chosen prompt, but *no generation* of text – i.e., the policy only receives the embeddings associated with the image and prompt (the left and middle red groupings of tokens at the top of Figure 2, but not the right-most group). This tests the hypothesis that representations of generated text might make certain task-relevant features more salient. For instance, the embeddings for "Spiders in Minecraft are black. Is there a spider in this image?", might not encode the presence of a spider as clearly as if the VLM generates "Yes" in response, impacting downstream performance. Finally, to check if our prompt evaluation and optimization strategy provides a good proxy for downstream task performance while tuning prompts for P2RL, we run PR2L with alternative prompts that were not predicted to be the best, as per our criterion in Appendix A. Concretely, this amounts to removing the auxiliary text from the prompt for *combat spider* and including it for *milk cow* and *shear sheep*.

Results from these ablation experiments are presented in Figure 5. In general, all of these ablations perform worse than PR2L. For *milk cow*, we note the most performant ablation is no generation, perhaps because the generated text is often wrong – among the chosen prompts, it yields the lowest true positive and negative rates for classifying the presence of its corresponding target entity (see Table 2 in Appendix A), though adding auxiliary text makes it even worse, perhaps explaining why *milk cow* experienced the largest performance decrease from adding it back in. Regardless, based on the overall trends, we conclude that (i) the *promptable* and *generative* aspects of VLM representations are important for extracting good features for control tasks and (ii) our simple evaluation scheme is an effective proxy measure of how good a prompt is for PR2L.

## 6 DISCUSSION

In this work, we propose Promptable Representations for Reinforcement Learning (PR2L), a method for extracting semantic features from images by prompting VLMs with task context, thereby making use of their extensive general-purpose prior knowledge. We demonstrate this approach in Minecraft, a domain that benefits from interpreting its visually-complex observations in terms of semantic concepts that can be related to task context. This general framework for using VLMs for control tasks opens many new paths of research. For example, prompts are currently hand-crafted based on the user's conception of useful features for the task. While coming up with effective prompts for our tasks in particular was not difficult, the process of generating and efficiently evaluating/optimizing them could be automated, which we leave for future works. Additionally, running PR2L with offline RL may provide even more in-depth insights into the benefits of this approach, since it removes the need for exploration (which we do not expect PR2L to help with). Finally, while we consider VLMs as our source of promptable representations, other types of promptable foundation models pre-trained with more sophisticated methods could also be used: e.g., ones trained on videos, domain-specific data, or even physical interactions might yield even better representations, perhaps which encode physics or action knowledge, rather than just common-sense visual semantics. Developing and using such models with PR2L offers an exciting way to transfer diverse prior knowledge to a broad range of control applications.

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

| Target Entity | Prompt | True Positive Rate | True Negative Rate |
|---|---|---|---|
| Spider | "Is there a spider in this image?" | 22.27% | 100.00% |
| | "Spiders in Minecraft are black. Is there a spider in this image?" | 73.42% | 94.54% |
| | "Spiders in Minecraft are black and have red eyes and long, thin legs. Is there a spider in this image?" | 50.50% | 99.85% |
| Cow | "Is there a cow in this image?" | 71.00% | 45.41% |
| | "Cows in Minecraft are black and white. Is there a cow in this image?" | 98.22% | 2.00% |
| | "Cows in Minecraft are black and white and have four legs. Is there a cow in this image?" | 96.67% | 7.35% |
| Sheep | "Is there a sheep in this image?" | 88.00% | 59.83% |
| | "Sheep in Minecraft are white. Is there a sheep in this image?" | 100.00% | 0.00% |
| | "Sheep in Minecraft are white and have four legs. Is there a sheep in this image?" | 100.00% | 0.00% |

Table 2: InstructBLIP's performance at decoding text indicating that it detected the presence of a target entity when given different prompts. We use this as a proxy metric for prompt engineering for RL, allowing us to determine which prompt to use for PR2L.

## A  PROMPT EVALUATION FOR RL IN MINECRAFT

We discuss how to evaluate prompts to use with PR2L, by showcasing an example for a Minecraft task. We start by noting that the presence and relative location of the entity of interest for each task (i.e., spiders, sheep, or cows) are good features for the policy to have. To evaluate if a prompt elicits these features from the VLM, we collect a small dataset of videos in which each Minecraft entity of interest is on the left, right, middle, or not on screen for the entirety of the clip. Each video is collected by a human player screen recording visual observations from Minecraft of the entity from different angles for around 30 seconds at 30 frames per second (with the exception of the video where the entity is not present, which is a minute long).

We propose prompts that target each of the two features we labeled. First, we evaluate prompts that ask "Is there a(n) [entity] in this image?" As the answers to these questions are just yes/no, we see how well the VLM can directly generate the correct answer for each frame in the collected videos. The VLM should answer "yes" for frames in the three videos where the target entity is on the left, right, or middle of the screen and "no" for the final video. Second, we evaluate if our prompts can extract the entity's relative position (left, right, or middle) in the videos where it is present. We note that the prompts we tried could not extract this feature in the decoded text (e.g., asking "Is the [entity] on the left, right, or middle of the screen?" will always cause the VLM to decode the same text). Thus, we try to see if this feature can be extracted from the decoded texts' representations. We measure this by fitting a three-category linear classifier of the entity's position given the *token-wise mean* of the decoded tokens' final embeddings. This is an unsophisticated and unexpressive classifier, i.e., we do not have to worry about the model potentially memorizing the data, which means that good classification performance corresponds to an easy extractability of said feature.

We evaluate three types of prompts per task entity for the first feature: one simply asking if the entity is present in the image (e.g., "Is there a spider in this image?") and two others adding varying amounts of auxiliary information about visual characteristics of the entity (e.g., "Spiders in Minecraft are black. Is there a spider in this image?" and "Spiders in Minecraft are black and have red eyes and long, thin legs. Is there a spider in this image?"). We present evaluations of all such prompts in Table 2. We find that the VLM benefits greatly from auxiliary information for the spider

case only, likely because spiders in Minecraft are the most dissimilar to the ones present in natural images of real spiders, whereas cows and sheep are still comparatively similar, especially in terms of scale and color. However, adding too much auxiliary information degrades performance, perhaps because the input prompt becomes too long, and therefore is out-of-distribution for the types of prompts that the VLM was pre-trained on. This same argument may explain why auxiliary information degrades performance for the other two target entities as well, causing them to almost always answer that said entities are present, even when they are not. Once more, considering that these targets exhibit a higher degree of visual resemblance to to their real counterparts compared to Minecraft spiders, it is reasonable to infer that the VLM would not benefit from auxiliary information. Furthermore, taking into account that the auxiliary information we gave is more common-sense than the information given for the spider, it could imply that the prompts are also more likely to be out-of-distribution (given that "sheep are white" is so obvious that people would not bother expressing it in language), causing the systematic performance degradation.

For the probing evaluation, we find that all three prompts reach similar final linear classifiabilities for each of their target entities, as shown in Figure 6. While this does not aid in choosing one prompt over another, it does confirm that the VLM's decoded embeddings for each prompt still contains this valuable and granular position information about the target entity, *even though the input prompt did not ask for it.*

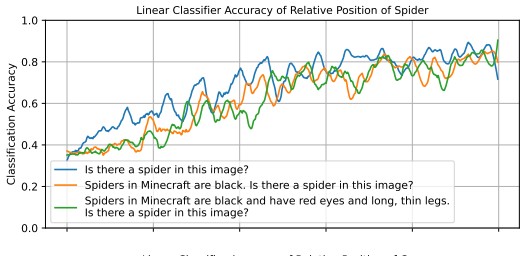

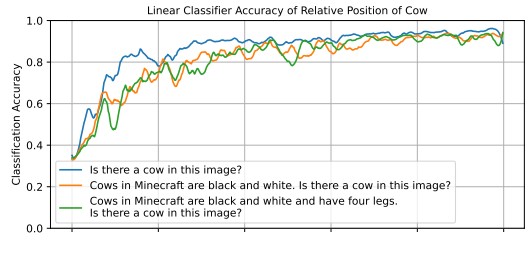

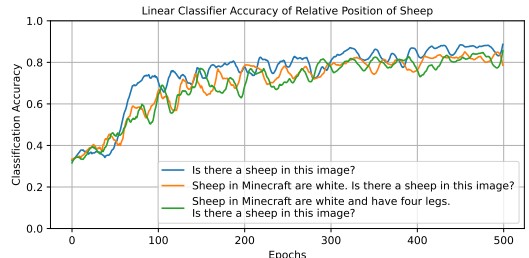

Figure 6: We train a linear classifier to predict the relative position of the target entity (left/right/middle) based on the average VLM embeddings decoded in response to each associated candidate prompt. We find that all three candidate prompts per task elicit embeddings that are similarly highly conducive to this classification scheme.

## B  MINEDOJO ENVIRONMENT DETAILS

**Spaces.** The observation space for the Minecraft tasks consists of the following:

1. **RGB:** Egocentric RGB images from the agent. (160, 256, 3)-size tensor of integers $\in \{0, 1, ..., 255\}$.

2. **Position:** Cartesian coordinates of agent in world frame. 3-element vector of floats.

3. **Pitch, Yaw:** Orientation of agent in world frame in degrees. Note that we limit the pitch to $15°$ above the horizon to $75°$ below for *combat spider*, which makes learning easier (as the agent otherwise often spends a significant amount of time looking straight up or down). Two 1-element vectors of floats.

4. **Previous Action:** The previous action taken by the agent. Set to no operation at the start of each episode. One-hot vector of size $|\mathcal{A}| = 53$ for *combat spider* and 89 otherwise (see below).

This differs from the simplified observation space used in Fan et al. (2022) in that we do not use any nearby voxel label information and impose pitch limits for *combat spider*. This observation space is the same for all Minecraft experiments.

The action space is discrete, consisting of 53 or 89 different actions:

1. **Turn:** Change the yaw and pitch of the agent. The yaw and pitch can be changed up to $\pm 90°$ in multiples of $15°$. As they can both be changed at the same time, there are

$9 \times 9 = 81$ total different turning actions. The turning action where the yaw and pitch changes are both $0°$ is the no operation action. Note that, since we impose pitch limits for the spider task, we also limit the change in pitch to $\pm 30°$, meaning there are only 45 turning actions in that case.

2. **Move:** Move forward, backward, left, right, jump up, or jump forward for 6 actions total.

3. **Attack:** Swing the held item at whatever is targeted at the center of the agent's view.

4. **Use Item:** Use the held item on whatever is targeted at the center of the agent's view. This is used to milk cows or shear sheep (with an empty bucket or shears respectively). If holding a sword and shield, this action will block attacks with the latter.

This non-*combat spider* action space is the same as the simplified one in Fan et al. (2022). All experiments for a given task share the same action space.

**World specifications.** MineDojo implements a fast reset functionality that we use. Instead of generating an entirely new world for each episode, fast reset simply respawns the player and all specified entities in the same world instance, but with fully restored items, health points, and other relevant task quantities. This lowers the time overhead of resets significantly, but also means that some changes to the world (like block destruction) are persistent. However, as breaking blocks generally takes multiple time steps of taking the same action (and does not directly lead to any reward), the agent empirically does not break many blocks aside from tall grass (which is destroyed with a single strike from any held item). We keep all reset parameters (like the agent respawn radius, how far away entities can spawn from the agent, etc) at their default values provided by MineDojo.

We stage all tasks in the same area of the same programmatically-generated world: namely, a sunflower plains biome in the world with seed 123. This is the default location for the implementation of the spider combat task in MineDojo. We choose this specific world/location as it represents a prototypical Minecraft scene with relatively easily-traversable terrain (thus making learning faster and easier).

**Additional task details and reward functions.** We provide additional notes about our Minecraft tasks.

*Combat spider*: Upon detecting the agent, the spider approaches and attacks; if the agent's health is depleted, then the episode terminates in failure. The agent receives $+1$ reward for striking any entity and $+10$ for defeating the spider. We also include several distractor animals (a cow, pig, chicken, and sheep) that passively wander the task space; the agent can reward game by striking these animals, making credit assignment of success rewards and the overall task harder.

*Milk cow*: The agent also holds wheat in its off hand, which causes the cow to approach the agent when detected and sufficiently nearby. For each episode, we track the minimum visually-observed distance between the agent and the cow at each time step. The agent receives $+0.1|\Delta d_{\min}|$ reward for decreasing this minimum distance (where $\Delta d_{\min} \leq 0$ is the change in this minimum distance at a given time step) and $+10$ for successfully milking the cow.

*Shear sheep*: As with *milk cow*, the agent holds wheat in its off hand to cause the sheep to approach it. The reward function also has the same structure as that task, albeit with different coefficients: $+|\Delta d_{\min}|$ for decreasing the minimum distance to the sheep and $+10$ for shearing it.

## C    POLICY AND TRAINING DETAILS

For our actual RL algorithm, we use the Stable-Baselines3 (version 2.0.0) implementation of clipping-based PPO (Raffin et al., 2021), with hyperparameters presented in Table 3. Many of these parameters are the same as the ones presented by Fan et al. (2022). For the spider trials, we use a cosine learning rate schedule:

$$\text{LR(current train step)} = \text{Min LR} + (\text{Max LR} - \text{Min LR}) \left( \frac{1 + \cos\left( \pi \frac{\text{current train step}}{\text{total train steps}} \right)}{2} \right) \quad (2)$$

We also present the policy and VLM hyperparameters in Table 4. The hyperparameters and architecture of the MLP part of the policy are primarily defined by the default values and structure

| Hyperparameter | Task | | |
|---|---|---|---|
| | *Combat Spider* | *Milk Cow* | *Shear Sheep* |
| Total Train Time Steps | 150000 | 100000 | 100000 |
| Rollout Steps | | 2048 | |
| Action Entropy Coefficient | | 5e-3 | |
| Value Function Coefficient | | 0.5 | |
| Max LR | 5e-5 | 1e-4 | 1e-4 |
| Min LR | 5e-6 | 1e-4 | 1e-4 |
| Batch Size | | 64 | |
| Update Epochs | | 10 | |
| $\gamma$ | | 0.99 | |
| GAE $\lambda$ | | 0.95 | |
| Clip Range | | 0.2 | |
| Max Gradient Norm | | 0.5 | |
| Normalize Advantages | | True | |

Table 3: PPO hyperparameters for Minecraft tasks, shared by the baselines, our method, and ablations.

| **Policy Transformer Hyperparameters** | |
|---|---|
| Transformer Token Size | 512 |
| Transformer Feedforward Dim | 512 |
| Transformer Number Heads | 2 |
| Transformer Number Decoder Layers | 1 |
| Transformer Number Encoder Layers | 1 |
| Transformer Output Dim | 128 |
| Transformer Dropout | 0.1 |
| Transformer Nonlinearity | ReLU |

| **Policy MLP Hyperparameters** | |
|---|---|
| Number Hidden Layers | 1 |
| Hidden Layer Size | 128 |
| Activation Function | tanh |

| **VLM Generation Hyperparameters** | |
|---|---|
| Max Tokens Generated | 6 |
| Min Tokens Generated | 6 |
| Decoding Scheme | Greedy |

Table 4: All policy hyperparameters for all Minecraft tasks.

defined by the Stable-Baselines3 `ActorCriticPolicy` class. Note that the no generation ablation, VLM image encoder baseline, and MineCLIP trials do not generate text with the VLM, and so all do not use the associated process's hyperparameters. The MineCLIP trials also do not use a Transformer layer in the policy, due to not receiving token sequence embeddings. It instead just uses a MLP, but with two hidden layers (to supplement the lowered policy expressivity due to the lack of a Transformer layer).

Additionally, InstructBLIP's token embeddings are larger than ViT-g/14's (used in the VLM image encoder baseline), and so may carry more information. However, the VLM does not receive any privileged information over the image encoder *from the task environment* – any additional information in the VLM's representations is therefore purely from the model's prompted internal knowledge. Still, to ensure consistent policy expressivity, we include a learned linear layer projecting all representations for this baseline and our approach to the same size (512 dimensions) so that the rest of the policy is the same for both.

## D  FULL ABLATION PLOTS

We extend Figure 5 with additional performance metrics in Figure 7.

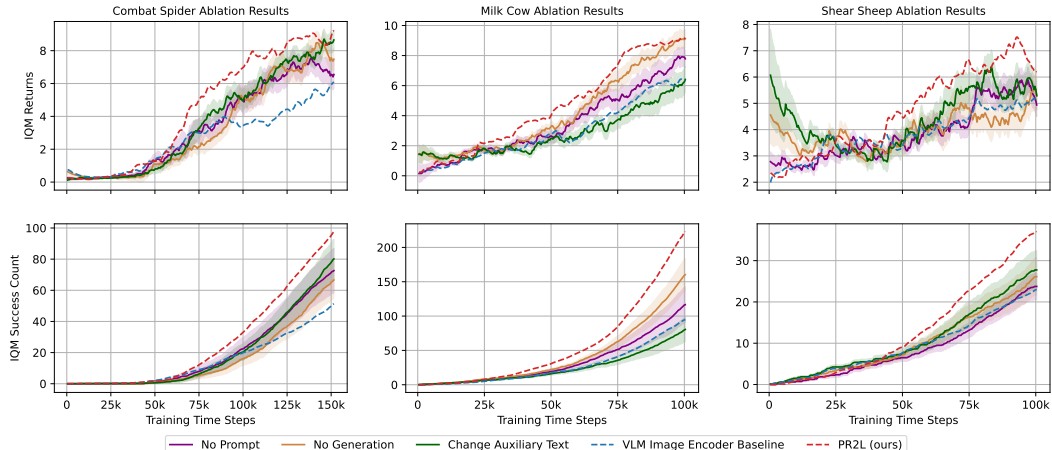

Figure 7: Extended version of Figure 5 with both the returns and success counts for the ablation trials. All curves represent IQMs and shaded regions represent the standard error **(though some are omitted for visual clarity)**.

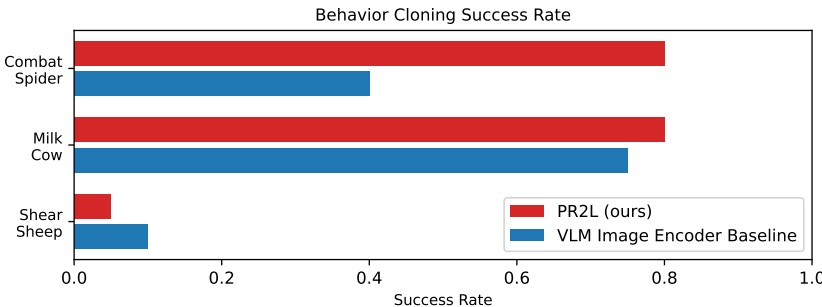

Figure 8: Success rates for BC on either PR2L or VLM image encoder baseline representations for all original tasks. PR2L excels at *combat spider*, even after the policy is trained for a single epoch.

## E    ADDITIONAL MINECRAFT EXPERIMENTS

### E.1    BEHAVIOR CLONING

We collected expert policy data by training a policy on MineCLIP embeddings to completion on all of our original tasks and saving all transitions to create an offline dataset. We then embedded all transitions with either PR2L or the VLM image encoder. Finally, we train policies with behavior cloning (BC) on successful trajectories under a specified length (300 for *combat spider*, 250 for *milk cow*, and 500 for *shear sheep*) from either set of embeddings for all three tasks, then evaluate their task success rates.

Results are presented in Fig. 8. We first note that, since the expert data was collected from a policy trained on MineCLIP embeddings, the *shear sheep* policy is not very effective (as we found in Fig. 3). Both resulting *shear sheep* BC policies are likewise not very performant. We find that *combat spider* in particular shows a very large gap in performance: the PR2L agent achieves approximately twice the success rate of the VLM image encoder agent *after training for just a single epoch*. The comparatively small amount of training and data necessary to achieve near-expert performance for this task supports our hypothesis that promptable representations from general-purpose VLMs do not help with exploration (they work better in offline cases, where exploration is not a problem), but instead are particularly conducive to being linked to appropriate actions even though the VLM is not producing actions itself. Further investigation of this hypothesis is presented in Appendix F.

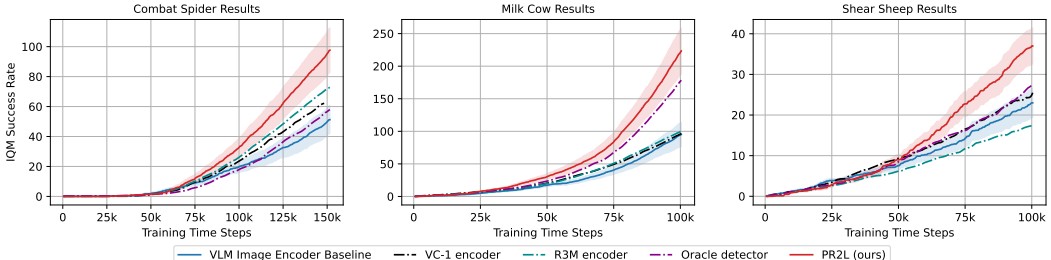

Figure 9: Success counts for the following additional baseline trials: (1) using VC-1 as an encoder, (2) using R3M as an encoder, and (3) using VLM image encoder embeddings with privileged oracle entity detection. All curves represent IQMs and shaded regions represent the standard error (some are omitted for visual clarity).

### E.2 ADDITIONAL BASELINES

We provide additional baselines in *combat spider*, *milk cow*, and *shear sheep*. Specifically, as InstructBLIP's image encoder may not be especially good for control tasks, we train policies on embeddings from VC-1 and R3M – two image encoders with representations that are specifically pretrained for embodied control and decision-making (Majumdar et al., 2023; Nair et al., 2022). As they both yield fix-sized embeddings, we use the same policy architecture and task hyperparameters as the MineCLIP baseline experiments. Additionally, to disambiguate whether PR2L is simply more performant due to being able to more reliably detect the task-relevant entity, we train a baseline policy on top of both the VLM image encoder embeddings and an indicator of whether the entity is in view based on a privileged oracle semantic LIDAR readings (as provided by MineDojo)[1]. This baseline uses the same hyperparameters as the VLM image encoder baseline. We present all results in Fig. 9 and find that PR2L beats all three baselines in all cases.

## F REPRESENTATION ANALYSIS

Why do our prompts yield higher performance than one asking for actions or instruction-following? Intuitively, despite appropriate responses to our task-relevant prompts not directly encoding actions, there should be a strong correlation: e.g., when fighting a spider, if the spider is in view and the VLM detects this, then a good policy should know to attack to get rewards. We therefore wish to investigate if our representations are conducive to easily deciding when certain rewarding actions would be appropriate for a given task – if it is, then such a policy may be more easily learned by RL, which would explain PR2L's improved performance over the baselines.

To investigate this, we use the embeddings of our offline data from the BC experiments (collected by training a MineCLIP encoder policy to high performance on all of our original three tasks, as discussed in Appendix E.1). We specifically look at the embeddings produced by a VLM when given our standard task-relevant prompts and when given the instructions used for our RT-2-style baseline. We then perform principal component analysis (PCA) on the tokenwise average of all embeddings for each observation, thereby projecting the embeddings to a 2D space with maximum variance.

We visualize these low-dimensional space in Fig. 10 for the final 20 successful observations from each task, with the point colors of orange and blue respectively indicating whether the observation results in a functional action (attack or use item) or movement (translation or rotation) by the expert policy. Additionally, we enlarge points corresponding to when the agent received rewards in order to recognize which actions aided in or achieved the task objective.

---

[1] We note that, as InstructBLIP uses its image encoder's representations as its sole source of visual information, if InstructBLIP *is* actually just doing better object detection, then any information about the presence of the entity must also be available to the VLM image encoder baseline.

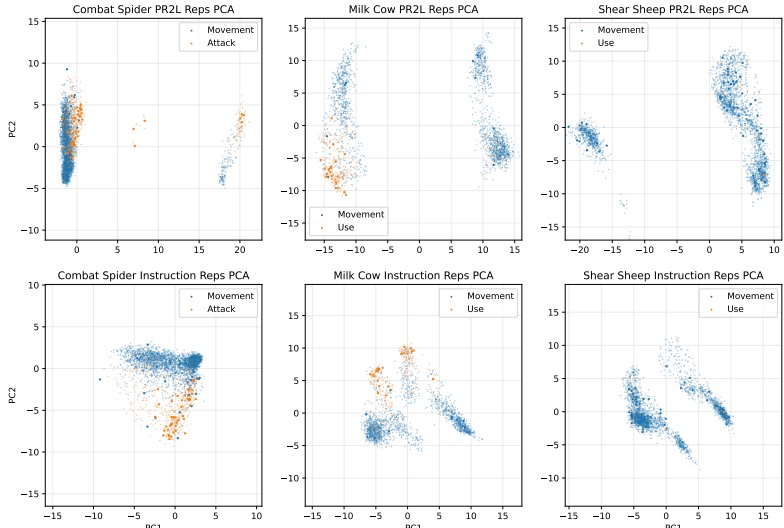

Figure 10: PCA of VLM representations of observations from twenty episode rollouts of expert policies in all three Minecraft tasks. Larger points correspond to transitions where the expert received > 0.1 reward. We vary the prompt to be either our task-relevant prompt or the RT-2-style baseline instruction prompt. Our prompt's representations are bi-modal, with the clusters on the left corresponding to the VLM outputting "yes" (the entity is in view). We find that most functional actions (orange points) that yielded rewards are located in said clusters. Note that, since these expert policies are trained on top of MineCLIP embeddings, the *shear sheep* policy is not very performant, as seen in Fig. 3.

We find that our considered prompts resulted in a bimodal distribution over representations, wherein the left-side cluster corresponds to the VLM outputting "yes (the entity is in view)" and the right-side one corresponds to "no." Additionally, observations resulting in functional actions that received rewards (large orange points in Fig. 10) tend to be on the left-side ("yes") cluster for representations elicited by our prompt, but are more widely distributed in the instruction prompt case, in agreement with intuition. This is especially clear in the *milk cow* plot, wherein nearly all rewarding functional actions (using the bucket on the cow to successfully collect milk) are in the lower left corner.

This analysis supports that the representations yielded by InstructBLIP in response to our chosen style of prompts are more structured than representations from instructions. Such structure is useful in identifying and learning rewarding actions, even when said actions were taken from an expert policy trained on unrelated embeddings. This suggests that such representations may similarly be more conducive to being mapped to good actions via RL, which we observe empirically (as our prompt's representations yield more performant policies than the instructions for the RT-2-style baseline).

## G EXTENDED LITERATURE REVIEW

**Learning in Minecraft.** We now consider some current approaches for creating autonomous learning systems for tasks in Minecraft. Such works highlight some of the difficulties prevalent in tasks in said environment. For instance, since Minecraft tasks take place in a dynamic open world, it can be difficult for an agent to determine what goal it is attempting to reach and how close it is to reaching that goal. Cai et al. (2023) tackles these issues by introducing and integrating a training scheme for self-supervised goal-conditioned representations and a horizon predictor. Zhou et al. (2023) learns a model from visual observations to discriminate between expert state sequences and non-expert ones, which provides a source of intrinsic rewards for downstream RL tasks (as it pushes the policy

to learn to match the expert state distribution, which tend to be "good" states for accomplishing tasks in Minecraft).

**Foundation Models and Minecraft.** Likewise, there has been much interest in applying foundation models – especially (V)LMs – to Minecraft tasks. Baker et al. (2022) pretrains on large scale videos, which enabled the first agent that could learn to acquire diamond tools (thereby completing a long-standing challenge in the MineRL competition Kanervisto et al. (2022)). LMs have subsequently also been used to produce graphs of proposed skills to learn or technology tree advancements to make in the form of structured language (Nottingham et al., 2023; Zhu et al., 2023; Yuan et al., 2023; Wang et al., 2023b). Other works propose to use the LLM to generate actions or code submodules given textual descriptions of observations or agent states (Wang et al., 2023a). Finally, VLMs have been used largely for language-conditioned reward shaping (Fan et al., 2022; Ding et al., 2023). In contrast, we use VLMs as a source of representations for learning of atomic tasks (as defined by Lin et al. (2023a)) that have pre-defined reward functions; the latter works can thus be used in conjunction with our proposed approach for tasks where these vision-language reward functions are appropriate.

