# OpenReview forum: "Vision-Language Models Provide Promptable Representations for Reinforcement Learning"
_ICLR.cc/2024/Conference — Submitted to ICLR 2024_

### Official Review · Reviewer_AR8Q · 2023-10-24

**Soundness:** 3 good
**Presentation:** 4 excellent
**Contribution:** 3 good
**Rating:** 5
**Confidence:** 5

**Summary:**

This work proposes a general task-related visual representation as input to the policy, aiming to improve the efficiency of reinforcement learning. By inputting task-related information (including auxiliary information) into VLM as prompts, efficient fusion with the current visual state is achieved. This paper verifies the effectiveness of this approach through experiments.

**Strengths:**

[1] The method proposed in this paper, PR2L, is very straightforward, and the story sounds reasonable. The organization of the paper is very clear, and it is polished well.

[2] The experimental analysis is quite comprehensive, and ablation experiments have demonstrated the effectiveness of each part of the proposed method. Conducting experiments in a challenging environment like Minecraft is persuasive.

**Weaknesses:**

[1] Concerns about computational cost. Obtaining promptable representation requires running a complex VLM on every image returned from the environment, along with the prompt and answer. This cost is prohibitively high and not practical.

[2] Concerns about the performance of the learned policy. The visually rich representation obtained at such a high computational cost should greatly improve the performance of the policy. However, the authors did not provide videos of rollouts on relevant tasks, making it difficult to judge the effectiveness of the promptable representation in a real Minecraft environment (the three tasks used in the paper are not very complex).

[3] Insufficient literature review. Since the authors conducted experiments in Minecraft, they should have provided a more comprehensive discussion of articles that control and plan within Minecraft. However, the authors left out the following important literature.

1. Open-world multi-task control through goal-aware representation learning and adaptive horizon prediction.
2. Video pretraining (vpt): Learning to act by watching unlabeled online videos.
3. Describe, explain, plan and select: Interactive planning with large language models enables open-world multi-task agents.
4. CLIP4MC: An RL-Friendly Vision-Language Model for Minecraft.
5. GROOT: Learning to Follow Instructions by Watching Gameplay Videos.
6. Learning from Visual Observation via Offline Pretrained State-to-Go Transformer.

**Questions:**

My main concerns are presented in the "Weaknesses" box, please refer to it.

---

> ### Author Response · Authors · 2023-11-18
>
> Thank you for your detailed feedback and for a positive assessment of our work. We have addressed your main concerns by adding the suggested related works, video rollouts and an explanation on why the computational cost of our method is actually much more practical than prior works that typically train entire VLMs. In the following we will answer your questions in detail.
>
> > Concerns about computational cost. Obtaining promptable representation requires running a complex VLM on every image returned from the environment, along with the prompt and answer. This cost is prohibitively high and not practical.
>
> We agree that the computational complexity of large VLMs is a major limiting factor for their adoption into control tasks and that lowering the compute required is difficult, barring architectural changes or lowered precision. Nevertheless, several prior works have used a large (V)LM in an RL loop [ https://arxiv.org/abs/2305.13301, https://llm-rl.github.io/static/images/llarp.pdf ] or even used a full pretrained VLM to initialize a policy (where the VLM is trained as part of the policy) [ https://arxiv.org/abs/2307.15818 ]. Compared to the latter, our method actually requires less compute, as it does not require training an entire VLM (only the lightweight policy network head).
> We also find that PR2L is reasonable in terms of wall-clock speed, achieving ~4 Hz inference speed (sufficient for playing Minecraft in real time). Training speed is also ~4 Hz, as gradients do not flow through the VLM (only the small policy network, which adds much less overhead). Moreover, training wall clock time is further improved in offline approaches, as all offline data can be embedded by VLMs in parallel and then reused in future epochs or runs. Again, this does not lower the compute required, but drastically decreases training time. We found this to be true in our new BC experiments (see Appendix E.1), wherein we trained near-expert level policies with comparatively small training time/data and used multiple VLMs to parallelize observation embedding generation.
>
> > Concerns about the performance of the learned policy. The visually rich representation obtained at such a high computational cost should greatly improve the performance of the policy. However, the authors did not provide videos of rollouts on relevant tasks, making it difficult to judge the effectiveness of the promptable representation in a real Minecraft environment (the three tasks used in the paper are not very complex).
>
> We include new videos visualizing 10 combat spider rollouts (here)[ https://drive.google.com/drive/folders/1GFN4C6qAEciGEK3phcqgYs3V-HxJpAOy?usp=sharing ] for both our approach and the image encoder baseline (achieving 7 and 4 successes respectively). We do note that it’s difficult to observe qualitative differences between the PR2L and baseline policies – they are more clear in reported aggregate performance metrics.
>
> > Insufficient literature review. Since the authors conducted experiments in Minecraft, they should have provided a more comprehensive discussion of articles that control and plan within Minecraft. However, the authors left out the following important literature.
>
> Thank you for these suggested citations – we have included them in the extended literature review in Appendix G.
>
> Please let us know if we have addressed your concerns, or if there is anything else we should consider. Otherwise, we would be extremely grateful if you could raise our score!

---

> > ### Comment · Reviewer_AR8Q · 2023-11-21
> >
> > Thanks for your responses. I appreciate the submitted rollout videos, it helps me to understand how the agent works. I agree that the proposed method is interesting and reasonable. I understand the authors have done a lot to speed up the training. However, the computational cost is still my major concern. 4Hz is too slow to scale up this method to large-scale tasks in Minecraft. After careful consideration, I decided to hold my original rating (5).

---

> ### Author Response · Authors · 2023-11-20
>
> Dear Reviewer AR8Q,
>
> We were wondering if you have had a chance to read our reply to your feedback. As the time window for the rebuttal is closing soon, please let us know if there are any additional questions we can answer to help raise the score!

---

> ### Author Response · Authors · 2023-11-21
>
> Thank you for your prompt response! We are glad that you find our method interesting. We would first like to clarify that our goal in this paper was to demonstrate our method PR2L can provide for an effective way to extract useful representations from pre-trained VLMs for downstream reinforcement learning. This goal realizes our long-term vision of using off-the-shelf pre-trained VLMs for RL, even though these models were never trained on data from the specific domain. As general-purpose VLMs start to become useful,  we believe it is crucial to develop approaches for extracting representations from them, targeted towards downstream RL, even though this comes at a larger inference cost with currently available VLMs. While we do agree that these computational costs can be reduced specifically for the domain of Minecraft, **we reiterate that the goal of this paper is not to develop better approaches for Minecraft, but to develop methodology to utilize pre-trained VLMs in RL.** Developing approaches for utilizing VLMs for downstream control has been an important theme in recent works (e.g., [RT-2](https://arxiv.org/abs/2307.15818), [InstructBLIP for Robotics](https://arxiv.org/pdf/2309.02561.pdf)), despite the associated computational costs.
>
> **Specifically, for our experiments, please note that a frequency of 4 Hz is already better than prior works that combine VLMs and RL:** [RT-2](https://arxiv.org/abs/2307.15818) uses a 65B parameter model and only achieves a worse 1-3 Hz control frequency on a real robot, which they still find to be sufficient for the real world.
>
> **Likewise, we use less compute and achieve greater sample efficiency than similar works:** As with our online RL experiments, [LLaRP](https://arxiv.org/abs/2310.17722)’s most performant model calls LLaMA-13B for 2e8 environment steps, thereby using a larger model (13B vs. our 7B parameters), with a longer history-dependent context length, for 1000 times more steps than PR2L. These prior approaches incur significantly larger computational costs, but are of interest to the community as they integrate foundation models with downstream RL / control algorithms.
>
> In addition, please note that when PR2L is applied with BC, as we demonstrate in our [newly added experiments](https://ibb.co/MRHrv03), the computational cost of encoding collected offline observations with the pre-trained VLM is incurred only once and can be parallelized. Encoding observations in the offline dataset takes ~2 hours with two VLMs (and only needs to be done once), which is comparable to running BC training itself (which takes around an hour for each run).
>
> **We would appreciate it if you are willing to upgrade your rating in the light of these clarifications. We are happy to discuss further. Thank you so much!**

---

### Official Review · Reviewer_Cm8x · 2023-10-28

**Soundness:** 3 good
**Presentation:** 3 good
**Contribution:** 2 fair
**Rating:** 5
**Confidence:** 4

**Summary:**

Paper presents an approach to provide task-relevant visual representations for RL, especially in an open-world environment.The main idea is to take a pre-trained VLM (vision-language model), feed it with the current visual observation, and a meticulously picked prompt about the current task, and then use the resulting embeddings produced by the last two layers of the VLM transformer as the representation. The authors also propose some additional techniques that could be helpful: 1) the VLM has to generate some text out of the prompt and use the embedding to correspond to both the prompt and the produced text, not just the embedding of the prompt only; 2) prompt engineering; 3) an encoder-decoder transformer is used as the policy to distill the representations into a summary embedding. Experiments and ablations on three Minecraft tasks show some promises.

**Strengths:**

+The study is relevant and could be of interest to many audiences with a background in large models, reinforcement learning, and representation learning.

+The method is well-motivated and technically sound. Pretrained VLM indeed provides open vocabulary and even knowledge-aided representations for multimodal input, which can be quite beneficial to open-world environments. Plus, it is plausible to tweak the representation further via prompting. This is a very neat idea.

+The results look impressive. Although the method is only evaluated on limited (3) tasks in a single environment (Minecraft), the advantages over the baselines and ablative approaches are significant. I do think the authors did a good job of comparing it against several interesting baselines, including no generation, no prompt, etc.

**Weaknesses:**

Having said those above, I do have some major concerns about the evaluation part of this paper. I also would like to point out some minor issues as well.

-Albeit the promises shown by the results on 3 tasks on Minecraft, I don't think the approach is thoroughly evaluated, especially given their claim on "leverage contextual prior information" and "visually-complex RL tasks" (see abstract). I have the following suggestions:

1) There are some other approaches that are designed to tackle similar issues, especially in Minecraft, ex. [1,2,3]. Although I agree some settings could be different (RL vs. IL, etc), they all deliver some backbone design or objective functions that could facilitate better representations. Comparisons against these missing baselines would help the reader with a better understanding of the significance of the proposed method.

2) Minecraft is indeed a challenging domain in terms of open-world and complex visual observations. However, the tasks being evaluated here (spider, cow, sheep) do not seem to be challenging enough to justify the effectiveness of the proposed method, especially on the claimed "leverage contextual prior information". These mobs are indeed very common and the tasks themselves do not seem to involve complex visual stimuli. My suggestion is to try some long-term and open-ended tasks like surviving, collecting items, etc. [7] offers a few of them and worth taking a look at.

Minor: some references on open-world representation learning and Minecraft agents should be cited: [1-6].


[1] open-world control: https://arxiv.org/abs/2301.10034

[2] VPT: https://arxiv.org/abs/2206.11795

[3] STG transformer: https://arxiv.org/abs/2306.12860

[4] DEPS: https://arxiv.org/abs/2302.01560

[5] Plan4MC: https://arxiv.org/abs/2303.16563

[6] GITM: https://arxiv.org/abs/2305.17144

[7] MCU: https://arxiv.org/abs/2310.08367

**Questions:**

-In Figure 2 and 3, why do some curves not have shadows?

-Some prompts shown in Table 1 require hand-crafted domain knowledge, ex. "Spiders in Minecraft are black". Is is possible to avoid this?

---

> ### Author Response · Authors · 2023-11-18
>
> Thank you for your detailed feedback and for a positive assessment of our work. We have addressed your main concerns by adding offline imitation learning experiments and more baselines (R3M, VC-1), to which our method favorably compares. In the following we will answer your questions in detail.
>
> > There are some other approaches that are designed to tackle similar issues, especially in Minecraft, ex. [1,2,3]. Although I agree some settings could be different (RL vs. IL, etc), they all deliver some backbone design or objective functions that could facilitate better representations. Comparisons against these missing baselines would help the reader with a better understanding of the significance of the proposed method.
>
> Thank you for the suggestion! We have run additional baselines (using VC-1 or R3M as encoders or giving the policy ground-truth entity detection) and have found PR2L outperforms all of them in all tasks. Additionally, we have also trained policies with BC on top of PR2L and image encoder representations, finding that PR2L especially excels at the “combat spider” task (doubling the baseline’s success rate and achieving near-expert performance in only one epoch).
>
> Additionally, our primary goal was not to build a SOTA system for Minecraft tasks, but to show that the representations produced by prompting a VLM with task context are better for learning control tasks than equivalent, non-prompted representations.
>
> In principle, our approach can be combined with other approaches for improving performance in Minecraft tasks (e.g., using a better reward function like MineCLIP or CLIP4MC). However, as we wanted to isolate performance differences to better representations elicited via prompting, we did not include such approaches.
>
> > Minecraft is indeed a challenging domain in terms of open-world and complex visual observations. However, the tasks being evaluated here (spider, cow, sheep) do not seem to be challenging enough to justify the effectiveness of the proposed method, especially on the claimed "leverage contextual prior information". These mobs are indeed very common and the tasks themselves do not seem to involve complex visual stimuli. My suggestion is to try some long-term and open-ended tasks like surviving, collecting items, etc. [7] offers a few of them and worth taking a look at.
>
> As we just wished to investigate the effectiveness of promptable representations (rather than developing an agent that is generally better at playing Minecraft), we chose to evaluate on the same task suite that the original MineDojo paper because they were solved by simple RL algorithms (off-the-shelf PPO) with little Minecraft-specific system engineering, meaning that our approach could likewise be less Minecraft-specific. We have likewise run PR2L and the VLM image encoder baseline on all five of the remaining programmatic tasks that MineDojo evaluated on to complete the analysis. For longer-horizon or more open-ended tasks, we likely would need to adopt similar approaches to the ones presented in your suggested citations.
>
> > Some prompts shown in Table 1 require hand-crafted domain knowledge, ex. "Spiders in Minecraft are black". Is it possible to avoid this?
>
> We find that the spider task’s domain knowledge is helpful, but not necessary. Removing it degrades performance slightly, but still outperforms using the non-promptable VLM image encoder representations (green vs. blue curves in Figs. 4 and 7’s combat spider plots). However, it does even better if this auxiliary text is included (red curve). We view the ability for VLMs to use auxiliary text as an advantage: it allows them to recognize stylized Minecraft spiders despite them being visually different from most natural images of spiders seen during pre-training. which intuitively explains why giving it this domain knowledge is helpful.
>
> Additionally, we recognize that re-running PR2L to optimize the prompt is computationally expensive. However, we find that the simple prompt evaluation scheme in Sec 5.2 and Appendix A is an effective proxy measure for how good a prompt is for PR2L.Finally, we also acknowledge that this evaluation requires domain-specific data, but note that it uses a miniscule amount of it compared to fully fine-tuning a model – we collected all prompt evaluation data in under an hour.
>
> [Continued in follow-up response due to character limit]

---

> > ### Author Response · Authors · 2023-11-18
> >
> > [Continuation from above]
> >
> > > In Figure 3 and 5, why do some curves not have shadows?
> >
> > Thank you for pointing this out: we omitted the shaded error region for the MineCLIP encoder trial in Fig. 3 and the PR2L/image encoder curves in Fig. 5 for visual clarity. We have updated the paper to state this. (Here)[https://ibb.co/Lk7J8W2] is a visualization of all error regions in Fig. 3 (in the original paper, only the green was missing). Note that the Fig. 5 red and blue curves are the same as the red and blue curves in Fig 3., which do have error regions.
> >
> > > Minor: some references on open-world representation learning and Minecraft agents should be cited: [1-6].
> >
> > Thank you for these suggested citations – we will include them in the extended literature review in Appendix G.
> >
> > Please let us know if we have addressed your concerns, or if there is anything else we should consider. Otherwise, we would be extremely grateful if you could raise our score!

---

> ### Author Response · Authors · 2023-11-20
>
> Dear Reviewer Cm8x,
>
> We were wondering if you have had a chance to read our reply to your feedback. As the time window for the rebuttal is closing soon, please let us know if there are any additional questions we can answer to help raise the score!

---

> > ### Comment · Reviewer_Cm8x · 2023-11-21
> > **Thank you for the reply**
> >
> > Thank you for the detailed reply. My concerns on baselines, handcrafted knowledge, plots, and citations are basically addressed.
> >
> > My concern about the task complexity remains. I can understand the need for evaluating tasks that can be solved by RL, but this seems to contradict your claim on "leverage contextual prior information", which I believe, requires evaluation on more challenging tasks.
> >
> > Imitation learning seems to be a better choice for more challenging tasks. Some other domains, ex. Atari, could be worth a try, as they also offer rich visual stimuli and likely can benefit from "contextual prior information".

---

> > > ### Author Response · Authors · 2023-11-22
> > >
> > > Dear Reviewer Cm8x,
> > >
> > > As a follow up to concern about "contextual prior information," we have started running preliminary experiments in the Habitat environment, so as to show the applicability of PR2L to non-Minecraft embodied control domains. We consider the ObjectNav task of learning to find toilets in a household environment and find that PR2L is able to achieve over 300 successes by the end of training, while the VLM image encoder baseline only achieves ~50 on average. See [here](https://ibb.co/7XB5MVn). The prompt we use is "Is a toilet likely in this room?" We confirm that it tends to answer "yes" to this question when shown images of bathrooms, even when a toilet is not directly in view, thereby demonstrating the use of VLM contextual knowledge for an application other than pure entity recognition (as shown in Minecraft). Here are examples of images the VLM answers ["yes"](https://drive.google.com/drive/folders/1EWG8mV-OL2oFZES23u9xFE9Y0d0u05O1?usp=sharing) and ["no"](https://drive.google.com/drive/folders/1DoJd5aSEUzL8Uz4ddqEkpB2Os9QHnz53?usp=sharing) to, respectively.
> > >
> > > These results are very preliminary, but we will be sure to include more extensive experiments in the final manuscript. We are running additional experiments now as well. We hope that such experiments and prompting methods better justify the "contextual prior information" claim. As the rebuttal period closes tonight, we would really appreciate if you let us know if you have any additional thoughts or feedback!

---

> ### Author Response · Authors · 2023-11-21
>
> Thank you for the prompt reply! We are glad we were able to address most of your concerns. We agree that there are more complex Minecraft tasks that could be tested on. Nevertheless, we find that the tasks we considered (all from the MineDojo paper) still pose an interesting and non-trivial challenge for reinforcement learning. In particular, we note that the original MineDojo paper found they needed a more complex learning algorithm than pure RL with PPO – namely, they require self-imitation learning to achieve good performance on shear sheep and milk cow (their approach fails otherwise, as shown in [Figure A.8](https://ibb.co/x2cSz7Q) of that paper).
>
> We empirically confirm that RL on top of MineCLIP does poorly on these tasks in particular, despite being trained to produce representations on domain-specific data (in contrast to our VLMs, which are not trained on Minecraft data). **Since these RL tasks are non-trivial even for more complex agent architectures/training algorithms that use domain-specific representations, we believe that our evaluation and contribution – showing a way to extract representations from _general purpose_ VLMs that match or significantly exceed the performance of domain-specific ones – is still valuable.**
>
> This level of performance is not present when using the VLM image encoder baseline, which suggests that the performance increase is because of the LLM parsing the image encoder’s representations based on its prior understanding of (1) what things look like and (2) how that relates to task context, provided as a prompt. **We show in our follow-up BC experiments that this results in representations that are particularly conducive to learning good actions** (with our combat spider agent achieving [expert-level performance after 1 epoch, doubling the success rate of the VLM image encoder baseline](https://ibb.co/MRHrv03)).
>
> An additional point regarding evaluating on other RL benchmarks such as Atari or dm control is that, as our method does not fine-tune a VLM on domain specific data (as prior works do), we are constrained to environments that produce somewhat more realistic observations, as the VLMs are trained on natural images.
>
> While we believe this all demonstrates how RL performance can be improved by using VLM representations that “leverage contextual prior information,” we would be happy to change the wording of that claim to something you believe to be a more accurate description of the above phenomenon – e.g., we could say that they “draw out relevant representations based on task context” (or some other suggested phrasing). Otherwise, if we have addressed your concern, we would be extremely grateful if you increase your rating to reflect this!

---

### Official Review · Reviewer_qGuw · 2023-10-29

**Soundness:** 3 good
**Presentation:** 3 good
**Contribution:** 2 fair
**Rating:** 6
**Confidence:** 4

**Summary:**

This paper introduces promptable representations for reinforcement learning (PR2L), which uses the semantic features of pre-trained VLM as state representation for reinforcement learning; the main advantage of PR2L to other pre-trained representations is that PR2L allows extract task-specific features from a generic pre-trained models by injecting task knowledge via prompting. PR2L outperforms both domain-specific representations and instruction-following methods on several tasks in MineCraft domain.

**Strengths:**

PR2L presents an interesting and creative way of utilizing pre-trained VLMs as representations for visual policy learning. It is unlike prior pre-trained representations for control work in which the features are generic (i.e., directly encoding the image observation); it is also different from recent Vision-Language-Action work (Brohan et al., 2023) in that it does not require fine-tuning a pre-trained VLM and enable high-frequency policies that are disentangled from the VLM backbone.

As VLMs are increasingly adopted in decision-making pipelines, PR2L is a timely work that presents a lightweight and simple alternative to the existing literature.

The paper itself is generally well-written and free of grammatical errors.

**Weaknesses:**

This paper's weaknesses mainly lie in its experimental methodologies.

1. The only form of prompt that PR2L uses essentially amounts to object detection in the scene. This introduces a confounding factor of whether PR2L outperforms baselines because it is able to recognize objects better in the scene.

2. The paper claims that the prompts are different from instruction following; however, the prompts are still manually constructed and task-specific. It's unclear the advantage of doing so as instructions, by construction, should exist as it is a direct form of task specification.

3. The improvements of PR2L over its various ablations appear only moderate. Furthermore, the best prompt format for the tasks are not consistent; for Spider, it is helpful to include contextual information of what a spider looks like in MineCraft, whereas for the other two tasks, it is more helpful to disregard such information. Therefore, applying PR2L to a new task may require substantial prompt engineering for the best performance.

4. PR2L is only evaluated on 3 tasks; these tasks are also the simplest in the MineDojo benchmark. The paper would be strengthened if more tasks and domains are evaluated. Currently, it is not convincing that PR2L can be generally applied to other visuomotor control domains. Relatedly, PR2L does not outperform MineCLIP on most tasks; given that MineCLIP exists and is open-sourced, PR2L's stated advantages can be better demonstrated via a new domain in which foundation pre-trained representations do not already exist.

5. BLIP-2's vision encoder may not be the strongest baseline for pre-trained vision encoders. Several prior works such as VC-1, R3M, MVP, VIP are trained for decision-making and robotics tasks and may constitute stronger baselines in that category.

**Questions:**

1. Could a baseline that somehow incorporates oracle object detection information be included? This will test whether PR2L does better because it detects the object of task interest in the scene.

2. Could more tasks and qualitatively different prompts be tested in the paper? Ideally, some tasks in MineCraft requires more than just object detection as auxillary information that may be implicitly captured by a VLM.

3. Could additional pre-trained vision encoder baselines be included?

I am willing to improve my assessment of the paper if these questions as well as the points in the Weakness section can be adequately addressed.

---

> ### Author Response · Authors · 2023-11-18
>
> Thank you for your detailed feedback. We have addressed your main concerns by visualizing the representations of instructions vs. prompts and running the suggested baselines with oracle object detection and running with image encoders (R3M, VC-1) that are more suited to embodied control, to which PR2L compares favorably. In the following we will answer your questions in detail.
>
> > The only form of prompt that PR2L uses essentially amounts to object detection in the scene. This introduces a confounding factor of whether PR2L outperforms baselines because it is able to recognize objects better in the scene.
> Could a baseline that somehow incorporates oracle object detection information be included? This will test whether PR2L does better because it detects the object of task interest in the scene.
>
> Thank you for the suggestion! To address the concern regarding object detection, we train baselines with oracle object detection and find that they perform worse than PR2L in all cases as seen [here](https://ibb.co/PrkhKFB) or Appendix E.2. Additionally, while we agree that our prompts essentially involve task-specific object detection, it's essential to highlight that this aligns with the specific requirements of our considered tasks. Detecting spiders, cows, and sheep in observations is a critical feature for accomplishing said tasks. The advantages of PR2L is that prompting (1) specifies what entities are important to detect for a task and (2) improves representations + detection of potentially OOD entities (like cartoony Minecraft spiders), obviating the need for domain-specific classifiers/representations by using general-purpose VLMs.
>
> > Relatedly, PR2L does not outperform MineCLIP on most tasks; given that MineCLIP exists and is open-sourced, PR2L's stated advantages can be better demonstrated via a new domain in which foundation pre-trained representations do not already exist.
> > BLIP-2's vision encoder may not be the strongest baseline for pre-trained vision encoders. Several prior works such as VC-1, R3M, MVP, VIP are trained for decision-making and robotics tasks and may constitute stronger baselines in that category.
> > Could additional pre-trained vision encoder baselines be included?
>
> Thank you for the suggestion, to address the concern regarding more baseline comparisons, we trained additional baselines on VC-1 and R3M and found that they perform worse than PR2L in all cases (see [the prior link](https://ibb.co/PrkhKFB)). We note that VC-1 is the most recent of these models, and seems to be more performant than MVP and VIP, so we train on its representations rather than the latter two.
>
> We chose to use InstructBLIP’s image encoder not because it was the best encoder for Minecraft, but to **illustrate that representations that are suboptimal for control are improved by passing them through an LLM with the task-relevant prompt**. The observed performance improvement of PR2L over the encoder baseline can thus entirely be attributed to the prompt extracting task-relevant information from the subpar generic encoder representations.
>
> > The paper claims that the prompts are different from instruction following; however, the prompts are still manually constructed and task-specific. It's unclear the advantage of doing so as instructions, by construction, should exist as it is a direct form of task specification.
>
> Thanks for bringing this up. We agree that instruction-following and our prompts both specify the task and require hand-designing. However, the former often cannot be done by our chosen VLM zero-shot, resulting in lower quality representations. This is reflected in the lower performance when the VLM is given instructions in our RT-2-style baseline.
>
> Furthermore, we find that the distribution of representations the VLM yields is very different depending on if it’s prompted with instructions or our approach, with the latter seemingly being more predictive of “good” actions to take. We observe this by plotting the first 2 principal components of the VLM features of observations from an expert policy when given instructions or our prompt, as shown [here](https://ibb.co/p1CL8Nx). We find that the representations from our prompt are bimodal (as the VLM usually answers “yes” or “no”) and most expert functional actions (attacking or using items) that result in rewards correspond to embeddings when the VLM answers “yes” (the left-hand clusters). This is most clear for milk cow – see the large orange points. In contrast, the PCA plots of instruction representations are much less structured; rewarding functional actions are much closer to the overall mean, and seem qualitatively harder to separate out and identify.
>
> **In summary: with our prompts, VLM representations of observations correspond strongly to actions that yield rewards for our considered tasks. This is not true of representations from instructions**. More details can be found in Appendix F.
>
> [Continued in follow-up response due to character limit]

---

> > ### Author Response · Authors · 2023-11-18
> >
> > [Continuation from above]
> >
> > > The improvements of PR2L over its various ablations appear only moderate. Furthermore, the best prompt format for the tasks are not consistent; for Spider, it is helpful to include contextual information of what a spider looks like in MineCraft, whereas for the other two tasks, it is more helpful to disregard such information. Therefore, applying PR2L to a new task may require substantial prompt engineering for the best performance.
> >
> > All ablations follow PR2L’s approach of using VLM representations for RL. Aside from the “no prompt” ablation, they are all given some form of task context. Thus, their improvement over the VLM image encoder baseline indicates that representations prompted with task context yield higher performance than non-prompted representations, albeit not as good as the main PR2L representations.
> >
> > The domain-specific auxiliary text is optional for combat spider. PR2L without it still significantly outperforms the pure image encoder baseline, as shown in ablations (green vs. blue curves in Figs. 5 and 7’s combat spider plots). However, it does even better if this auxiliary text is included (red curve). As discussed in Sec. 5.1, we view the ability for VLMs to use auxiliary text as an advantage: it allows them to recognize stylized Minecraft spiders despite them being visually different from most natural images of spiders seen during pre-training.
> >
> > Additionally, we recognize that re-running PR2L to optimize the prompt is computationally expensive. However, we find that the simple prompt evaluation scheme in Sec 5.2 and Appendix A is an effective proxy measure for how good a prompt is for PR2L. In particular, it seems like the auxiliary information for the cow / sheep tasks is detrimental because it causes the VLM’s outputs to degrade: it always answers “yes” to “is there a cow / sheep in this image?” when the auxiliary text is added (Table 2). This may be fixed by using a more advanced VLM, but for now, our evaluation approach is able to catch this systematic bias in response to auxiliary text. Finally, we also acknowledge that this evaluation requires domain-specific data, but note that it uses a miniscule amount of it compared to fully fine-tuning a model – we collected all prompt evaluation data in under an hour.
> >
> > > PR2L is only evaluated on 3 tasks; these tasks are also the simplest in the MineDojo benchmark. The paper would be strengthened if more tasks and domains are evaluated. Currently, it is not convincing that PR2L can be generally applied to other visuomotor control domains.
> >
> > Thank you for the suggestion. The MineDojo paper evaluated their agent on 8 “programmatic” tasks (and 4 of which were “creative,” and thus do not have associated concrete reward functions). We thus ran additional experiments comparing PR2L and the VLM image encoder baseline on the remaining 5 programmatic tasks. Only “combat zombie” is currently available (as our computing cluster’s file system is down), but we find that PR2L outperforms the baseline in that case, as shown [here](https://ibb.co/jggxzD9).
> >
> > Please let us know if we have addressed your concerns, or if there is anything else we should consider. Otherwise, we would be extremely grateful if you could raise our score!

---

> ### Author Response · Authors · 2023-11-20
>
> Dear Reviewer qGuw,
>
> We were wondering if you have had a chance to read our reply to your feedback. As the time window for the rebuttal is closing soon, please let us know if there are any additional questions we can answer to help raise the score!

---

> ### Author Response · Authors · 2023-11-21
>
> Dear Reviewer qGuw,
>
> As the rebuttal window closes tomorrow, we would again like to ask if you have had a chance to consider our reply to your feedback. Please let us know if you have any additional concerns we can address to help raise the score!

---

> > ### Comment · Reviewer_qGuw · 2023-11-22
> >
> > Dear Authors,
> >
> > Thank you for your responses and I have carefully read through them. Most of my concerns have been adequately address except the concern over "the best prompt format for the tasks are not consistent" and that only one additional environment has been added. In addition, it's not entirely clearly to me why the benefit of the proposed approach appears to show only later during training. That said, despite the room for improvement for the empirical results and their analysis, I do think that the paper has been strengthened and presents an interesting and novel idea that can be explored further in research, so I have decided to increase my score.

---

> > > ### Author Response · Authors · 2023-11-22
> > >
> > > We are very happy that we were able to address your concerns and are very grateful for the score increase! Regarding your current concerns:
> > >
> > > > It's not entirely clear to me why the benefit of the proposed approach appears to show only later during training.
> > >
> > > Thank you for bringing this up – We hypothesize that this is because PR2L does _not_ necessarily aid in exploration, but instead because the VLM produces representations of observations that are more conducive to being linked to rewards and good actions than ones from the VLM image encoder. For instance, in the milk cow task, the policy should learn that when the cow is in view, then walking up to it and using the bucket results in rewards and success. It is much more obvious to the policy that the cow is in view when using PR2L, since the VLM directly produces distinct embeddings corresponding to answering “yes” to “is there a cow in this image?”, rather than having to learn to extract this pattern from context-less visual embeddings.
> > >
> > > Nevertheless, while this empirically makes learning good actions easier with PR2L, it does not necessarily aid in online exploration needed to actually discover those actions in the first place. This suggests that offline RL or behavior cloning (where exploration is not a problem) are even better suited for leveraging PR2L. The results of our behavior cloning experiments support this fact: for combat spider, we are able to train the PR2L policy to expert level performance, doubling the VLM image encoder policy’s performance with just one epoch of training (see [here](https://ibb.co/MRHrv03) or Appendix E.1 for more details).
> > >
> > > We will be sure to clarify this in the final version of our manuscript.
> > >
> > > > only one additional environment has been added
> > >
> > > Unfortunately, our computing cluster issues have prevented us from adding the full set of additional experiments. However, we have started running preliminary experiments in the Habitat environment, so as to show the applicability of PR2L to non-Minecraft embodied control domains. We consider the ObjectNav task of learning to find toilets in a household environment and find that PR2L is able to achieve over 300 successes by the end of training, while the VLM image encoder baseline only achieves ~50 on average. See [here](https://ibb.co/7XB5MVn). The prompt we use is “Is a toilet likely in this room?” We confirm that it tends to answer “yes” to this question when shown images of bathrooms, even when a toilet is not directly in view, thereby demonstrating the use of VLM contextual knowledge for an application other than pure entity recognition (as shown in Minecraft). Here are examples of images the VLM answers [“yes”](https://drive.google.com/drive/folders/1EWG8mV-OL2oFZES23u9xFE9Y0d0u05O1?usp=sharing) and [“no”](https://drive.google.com/drive/folders/1DoJd5aSEUzL8Uz4ddqEkpB2Os9QHnz53?usp=sharing) to, respectively.
> > >
> > > These results are very preliminary, but we will be sure to include more extensive experiments in the final manuscript. We are running additional experiments now as well, and will update if additional results are available before the rebuttal period ends.
> > >
> > > > "the best prompt format for the tasks are not consistent"
> > >
> > > We are not sure if you got a chance to read this, but we tried to address this question previously in our answer (see below). Is there any specific question you would like us to address or do you have any concern with said answer we could help with?
> > >
> > > > The domain-specific auxiliary text is optional for combat spider. PR2L without it still significantly outperforms the pure image encoder baseline, as shown in ablations (green vs. blue curves in Figs. 5 and 7’s combat spider plots). However, it does even better if this auxiliary text is included (red curve). As discussed in Sec. 5.1, we view the ability for VLMs to use auxiliary text as an advantage: it allows them to recognize stylized Minecraft spiders despite them being visually different from most natural images of spiders seen during pre-training.
> > >
> > > > Additionally, we recognize that re-running PR2L to optimize the prompt is computationally expensive. However, we find that the simple prompt evaluation scheme in Sec 5.2 and Appendix A is an effective proxy measure for how good a prompt is for PR2L. In particular, it seems like the auxiliary information for the cow / sheep tasks is detrimental because it causes the VLM’s outputs to degrade: it always answers “yes” to “is there a cow / sheep in this image?” when the auxiliary text is added (Table 2). This may be fixed by using a more advanced VLM, but for now, our evaluation approach is able to catch this systematic bias in response to auxiliary text. Finally, we also acknowledge that this evaluation requires domain-specific data, but note that it uses a miniscule amount of it compared to fully fine-tuning a model – we collected all prompt evaluation data in under an hour.

---

### Official Review · Reviewer_sWjV · 2023-11-01

**Soundness:** 2 fair
**Presentation:** 4 excellent
**Contribution:** 3 good
**Rating:** 6
**Confidence:** 4

**Summary:**

The paper presents a simple but effective approach of improving visual embeddings from pre-trained VLMs by providing VLMs with useful prompts. For example, to detect spider in Minecraft, instead of directly encoding the scene using a vision encoder, the authors extract more meaningful representations from the VLM by giving auxiliary information like “Spiders in Minecraft are black. Is there a spider in the image?”. This can help the VLM to focus on task-specific/domain-specific information and produce more meaningful embeddings that are useful for training a RL policy. The authors show that the proposed approach outperforms policies trained using generic image features from a vision encoder ( on 3/3 tasks), as well as domain specific image features (on 2/3 tasks).

**Strengths:**

- The main contribution of the paper “Prompting VLM via auxiliary information and task context” allows extracting more meaningful representation from a VLM is quite interesting and easily applicable to a range of tasks. Instead of fine-tuning VLM for specific domains, it’s easier to plug-and-play existing VLM and extract meaningful representations via prompting.
- Overall, the paper is well written and is systematic in its evaluation. I also appreciate the authors willingness to address concerns preemptively (lack of visual tokens as input to the policy MLP, not fine-tuning VLM similar to RT-1).

**Weaknesses:**

- Given that the approach is using a VLM, it’d be nice to test the model for “unseen” tasks, containing objects and instructions not seen during training. For instance, does the policy generalise form “Combat Spider” to “Combat Zombie”?
- I also recommend a stronger evaluation on Minecraft benchmark consistent with the evaluations done in MineDOJO. Currently, the paper shows result on only three tasks. For a more exhaustive evaluation, MineDOJO recommends evaluation on a collection of starter tasks (32 programmatic and 32 creative tasks).
- While I understand that the authors didn’t have the resources to train a RT-1 style baseline, would it still be possible to train an action decoder on top of the VLM to produce actions. I think having a strong RT-1 style baseline is very important to properly evaluate the question (2) mentioned in the paper — “How does PR2L compare to approaches that directly “ask” the VLM to generate the best possible actions for a task specified in the prompt?”

Overall, I liked the main contribution of the paper. But I believe, in its current form, the evaluation in the paper is a bit weak and could be made more exhaustive (unseen tasks, more exhaustive MineCraft evaluation).


**Update**: After reading the rebuttal, most of my questions are adequately answered. I still believe that directly training an action decoder (while keeping VLM frozen is a good baseline) and should be included in the paper. I also feel that the paper will be stronger if they include more exhaustive minecraft experiments (to check for generalisation on unseen tasks / objects) and more environments which are visually more complex like Habitat / AI2 Thor. Based on authors response, I am increasing my score.

**Questions:**

Apart from questions asked in the weaknesses section, I have additional questions:

- The proposed architecture compresses the task-relevant features from the VLM into a single CLS token which can severely restrict the information available to the policy. While this may work for simpler environments like Minecraft which doesn’t have a lot of clutter, it might not work for other environments / tasks (rearrangement tasks in indoor environments). Did the authors try an approach like Perceiver IO (Jaegle et al, 2021) which encodes N tokens to K (1≤K≤N) tokens?
- I think the first question — “Can promptable representations obtained via task-specific prompts enable more efficient learning than those of pretrained image encoders?” is not really answered. It’s unclear what efficiency mean (faster to train in FLOPS? faster to train measured by amount of training steps?). I think the paper can be made stronger by comparing training efficiency when using the proposed approach vs using VLM image-representations directly.
- I didn’t fully understand the various ablations done. Specifically, did the authors try just giving task context (and no auxiliary information)? Similarly, did the authors try giving just auxiliary information without giving task context.
- While I don’t expect this experiment to be performed for rebuttal, I really wish the authors evaluated their approach on tasks that are visually more complex (or the environments are more cluttered). For [e.g](http://e.gm)., using the approach to perform rearrangement tasks in indoor environments like Habitat / AI2 Thor.

---

> ### Author Response · Authors · 2023-11-18
>
> Thank you for your detailed feedback. We have addressed your main concerns by adding all remaining programmatic Minecraft tasks from the original MineDojo paper and three additional baselines to which our method favorably compares. In the following we will answer your questions in detail.
>
> > I also recommend a stronger evaluation on the Minecraft benchmark consistent with the evaluations done in MineDOJO. Currently, the paper shows results on only three tasks. For a more exhaustive evaluation, MineDOJO recommends evaluation on a collection of starter tasks (32 programmatic and 32 creative tasks).
>
> Thank you for the suggestion! Unfortunately, to the best of our knowledge, we were unable to find the 64 core tasks mentioned in the original MineDojo paper. As the MineDojo paper evaluates their agent on only 12 tasks (four of which were “creative,” and thus do not have associated concrete reward functions), we assume that said tasks are part of the core set. Thus, we run PR2L and the VLM image encoder baseline on all 8 of the “programmatic” tasks considered in the MineDojo paper. Only “combat zombie” is currently available (as our computing cluster’s file system is down), but we find that PR2L outperforms the baseline in that case, as shown [here](https://ibb.co/jggxzD9).
> We have also run some additional baselines / ablations on the original three tasks, wherein the model is trained on VC-1 and R3M representations or is supplemented by ground-truth entity detection. In all cases, these perform worse than our approach, as shown [here](https://ibb.co/PrkhKFB) and discussed in Appendix E.2.
>
> > While I understand that the authors didn’t have the resources to train a RT-1 style baseline, would it still be possible to train an action decoder on top of the VLM to produce actions. I think having a strong RT-1 style baseline is very important to properly evaluate the question (2) mentioned in the paper — “How does PR2L compare to approaches that directly “ask” the VLM to generate the best possible actions for a task specified in the prompt?”
>
> Great point! While we did not exactly match the architecture of RT-1/-2, we already addressed question (2) in our baseline (b), wherein the VLM was given a prompt similar to the ones used in the RT-1 and RT-2  with the task instruction and possible actions, and was then asked to produce an action. The trained Transformer-based policy then decodes the instructions and generated text into actions. Of course, unlike RT-2, this approach does not fine-tune the VLM itself on data from MineCraft, but please note that none of our methods use a fine-tuned VLM: our goal is to compare different approaches to query representations from a pre-trained model in terms of their downstream RL performance. We apologize for any confusion and have updated the paper to clarify this point.
>
> > The proposed architecture compresses the task-relevant features from the VLM into a single CLS token which can severely restrict the information available to the policy ... Did the authors try an approach like Perceiver IO (Jaegle et al, 2021) which encodes N tokens to K (1≤K≤N) tokens?
>
> For a fair comparison against MineCLIP that produces a fixed-length representation of an observation,  we chose to convert the VLM representations to a single-token bottleneck. This allows us to roughly match the size of the policy network that converts the VLM representation into an action distribution. In our experiments, we do not find this bottleneck to hinder our policy’s ability to achieve high task performance. That said, of course, PR2L can use more advanced policy architectures which do not involve such bottlenecks and we would only expect these architectures to improve performance.
>
> > It’s unclear what efficiency means (faster to train in FLOPS? faster to train measured by amount of training steps?). I think the paper can be made stronger by comparing training efficiency when using the proposed approach vs using VLM image-representations directly.
>
> Thank you for this suggestion. We apologize for the confusion – we use the term efficiency to denote sample efficiency, following the standard textbook definitions of sample efficiency in RL. Sample efficiency is distinct from computation efficiency (measured in FLOPs), and is a standard comparison metric for RL algorithms. We have updated the paper to reflect this. The reviewer-suggested comparison is shown in the red (PR2L) and blue (VLM image representation) curves in all performance figures – when considering sample efficiency, we can simply check the returns or successes after a fixed number of training samples.
>
> [Continued in follow-up response due to character limit]

---

> > ### Author Response · Authors · 2023-11-18
> >
> > [Continuation from above]
> > > I didn’t fully understand the various ablations done. Specifically, did the authors try just giving task context (and no auxiliary information)? Similarly, did the authors try giving just auxiliary information without giving task context.
> >
> > We apologize for the confusion – our ablations considered (1) removing the prompt altogether, (2) removing the VLM’s text generation abilities, and (3) adding auxiliary text if our main trial did not have it and vice-versa (task context is always there). Ablations (1) and (2) correspond to removing the middle and right-most groupings of red embeddings shown in Fig. 2 respectively. For the spider task (which has auxiliary text), (3) amounts to only providing task context (green curve in Fig. 5). The main prompts for the other tasks already only have task context (red curves in Figs. 3 and 5).
> >
> > > While I don’t expect this experiment to be performed for rebuttal, I really wish the authors evaluated their approach on tasks that are visually more complex (or the environments are more cluttered). For e.g., using the approach to perform rearrangement tasks in indoor environments like Habitat / AI2 Thor.
> >
> > Thank you for the suggested additional task environments. While we have developed infrastructure to run experiments in Habitat, we ran into technical difficulties when running experiments, but are working to resolve that now.
> >
> > Please let us know if we have addressed your concerns, or if there is anything else we should consider. Otherwise, we would be extremely grateful if you could raise our score!

---

> > ### Author Response · Authors · 2023-11-22
> >
> > Dear Reviewer sWjV,
> >
> > We wanted to remind you that the rebuttal window closes tonight, so we would be extremely grateful for any feedback on our response!
> >
> > Additionally, as you suggested and as mentioned in our initial reply, we have started running preliminary experiments in the Habitat environment, so as to show the applicability of PR2L to non-Minecraft embodied control domains. We consider the ObjectNav task of learning to find toilets in a household environment and find that PR2L is able to achieve over 300 successes by the end of training, while the VLM image encoder baseline only achieves ~50 on average. See [here](https://ibb.co/7XB5MVn). The prompt we use is "Is a toilet likely in this room?" We confirm that it tends to answer "yes" to this question when shown images of bathrooms, even when a toilet is not directly in view, thereby demonstrating the use of VLM contextual knowledge for an application other than pure entity recognition (as shown in Minecraft). Here are examples of images the VLM answers ["yes"](https://drive.google.com/drive/folders/1EWG8mV-OL2oFZES23u9xFE9Y0d0u05O1?usp=sharing) and ["no"](https://drive.google.com/drive/folders/1DoJd5aSEUzL8Uz4ddqEkpB2Os9QHnz53?usp=sharing) to, respectively.
> >
> > These results are very preliminary, but we will be sure to include more extensive experiments in the final manuscript. We are running additional experiments now as well.

---

> ### Author Response · Authors · 2023-11-20
>
> Dear Reviewer sWjV,
>
> We were wondering if you have had a chance to read our reply to your feedback. As the time window for the rebuttal is closing soon, please let us know if there are any additional questions we can answer to help raise the score!

---

> ### Author Response · Authors · 2023-11-21
>
> Dear Reviewer sWjV,
>
> As the rebuttal window closes tomorrow, we would again like to ask if you have had a chance to consider our reply to your feedback. Please let us know if you have any additional concerns we can address to help raise the score!

---

### Author Response · Authors · 2023-11-18

We thank all reviewers for their thoughtful and detailed comments! We first provide some general updates and result summaries:

- Most reviewers suggested evaluating on a) more tasks b) more baselines. To address this, we have now run PR2L and the VLM image encoder baseline on the five programmatic tasks from the original MineDojo paper which we did not initially consider: combat zombie, hunt cow, hunt sheep, combat enderman, and combat pigman. We thus have run experiments on all of the tasks which said paper trained policies for (minus the four “creative” tasks, which lack concrete reward functions). For combat zombie, we find that PR2L does much better than the VLM image encoder baseline – see [here](https://ibb.co/jggxzD9). Unfortunately, results from the rest of the tasks are currently unavailable due to unforeseen technical difficulties on our computing server – specifically, our cluster file system went down, so we have been unable to access our latest experimental results. We will post an update as soon as these logs are accessible again.
- Additionally, we also have added reviewer-proposed baselines (policies trained on VC-1, R3M, and oracle object detection), ultimately finding that PR2L outperforms all these approaches on our original tasks (see [here](https://ibb.co/PrkhKFB) or Appendix E.2 for more details).
- We also show that our method yields good results for the offline case with a behavioral cloning experiment (see [here](https://ibb.co/MRHrv03) or Appendix E.1 for more details) – especially in the best case of combat spider, wherein the PR2L BC policy doubles the performance of the VLM image encoder baseline after just one epoch of training.
- We appreciate all suggested citations and have added an extended literature review to Appendix G that focuses more on Minecraft-specific works.

We have also responded to address each reviewer’s concerns individually below.

---

### Author Response · Authors · 2023-11-22

Dear all reviewers,

We have started running preliminary experiments in the Habitat environment, so as to show the applicability of PR2L to non-Minecraft embodied control domains. We consider the ObjectNav task of learning to find toilets in a household environment and find that PR2L is able to achieve over 300 successes by the end of training, while the VLM image encoder baseline only achieves ~50 on average. See [here](https://ibb.co/7XB5MVn). The prompt we use is "Is a toilet likely in this room?" We confirm that it tends to answer "yes" to this question when shown images of bathrooms, even when a toilet is not directly in view, thereby demonstrating the use of VLM contextual knowledge for an application other than pure entity recognition (as shown in Minecraft). Here are examples of images the VLM answers ["yes"](https://drive.google.com/drive/folders/1EWG8mV-OL2oFZES23u9xFE9Y0d0u05O1?usp=sharing) and ["no"](https://drive.google.com/drive/folders/1DoJd5aSEUzL8Uz4ddqEkpB2Os9QHnz53?usp=sharing) to, respectively.

These results are very preliminary, but we will be sure to include more extensive experiments in the final manuscript. We are running additional experiments now as well.

---

### Meta-Review · Area_Chair_wpJF · 2023-12-03

**Metareview:**

The paper received borderline ratings from the reviewers (6, 6, 5, 5), highlighting several concerns such as:
- Need for a stronger evaluation benchmark,
- Limitations in prompts confined to object detection,
- Moderate improvements over ablations,
- A call for stronger VLM baselines,
- Insufficient literature review,
- and many other issues.

The rebuttal successfully addressed some concerns, resulting in a score increase from some of the reviewers. The AC checked the paper, the reviews, the rebuttal, and the responses. While the AC appreciates the simplicity of the idea, the evaluation benchmark is not strong. The main issue is that the three tasks chosen for evaluation are quite simplistic. The rebuttal introduces new experiments on other tasks and the Habitat environment, but they remain unfinished or yield very preliminary results. This incompleteness renders the paper lacking. Coupled with other concerns raised by the reviewers, it led the AC to the decision to reject the paper. The AC acknowledges technical difficulties on the computing cluster, but the paper should have included these results in the first place. The AC encourages the authors to revise the paper, include the additional experiments, and submit it to a future venue.

**Justification For Why Not Higher Score:**

The paper should employ stronger evaluation benchmarks, as the current tasks are simplistic. Additionally, the power of VLMs lies in their open-vocabulary nature; therefore, exploration of generalization to unseen tasks and objects should have been conducted.

**Justification For Why Not Lower Score:**

N/A

---

### Decision · Program_Chairs · 2024-01-16

Reject